# PRIVATE-RAG: ANSWERING MULTIPLE QUERIES WITH LLMS WHILE KEEPING YOUR DATA PRIVATE

## ABSTRACT

Retrieval-augmented generation (RAG) enhances large language models (LLMs) by retrieving documents from an external corpus at inference time. When this corpus contains sensitive information, however, unprotected RAG systems are at risk of leaking private information. Prior work has introduced differential privacy (DP) guarantees for RAG, but only in single-query settings, which fall short of realistic usage. In this paper, we study the more practical multi-query setting and propose two DP-RAG algorithms. The first, MURAG, leverages an individual privacy filter so that the accumulated privacy loss only depends on how frequently each document is retrieved rather than the total number of queries. The second, MURAG-ADA, further improves utility by privately releasing query-specific thresholds, enabling more precise selection of relevant documents. Our experiments across multiple LLMs and datasets demonstrate that the proposed methods scale to hundreds of queries within a practical DP budget ($\varepsilon \approx 10$), while preserving meaningful utility.

## 1 INTRODUCTION

Retrieval-augmented generation (RAG) has become a popular approach for deploying large language models (LLMs) in real-world applications. A core feature of RAG is its reliance on an external dataset as the primary knowledge source at inference time. For example, a medical RAG system may retrieve historical patient records to answer clinical questions more accurately. However, such external datasets often contain sensitive or confidential information. In domains like healthcare or law, the retrieved content may expose private records, raising serious privacy concerns. Prior work has shown that RAG systems without proper safeguards are vulnerable to information leakage (Naseh et al., 2025; Liu et al., 2025; Anderson et al., 2024; Li et al., 2025; Zhang et al., 2025; Zeng et al., 2024a; Jiang et al., 2024; Peng et al., 2024), compromising data owner privacy and user trust.

Differential privacy (DP) is a widely adopted framework for providing rigorous guarantees on individual data protection. Recent work (Koga et al., 2024) has proposed DPSparseVoteRAG, a RAG system that ensures the generated answer satisfies DP with respect to the external dataset, for *a single user query*. Empirical results demonstrate that this approach outperforms the baseline using a public LLM without the external dataset, while achieving an $\varepsilon$-DP guarantee with $\varepsilon \approx 10$.

In realistic deployments, many queries may be issued by one or more users. A naïve approach that applies DPSparseVoteRAG to each query and relies on standard composition theorems quickly exhausts a reasonable privacy budget. As our experimental results (Figure 2) show, to achieve reasonable utility, this approach may require a privacy budget as large as $\varepsilon = 1000$, which is generally considered too weak. This raises a key question:

*Can we design a differentially private RAG algorithm that handles hundreds of queries while ensuring both meaningful privacy and utility?*

We answer this question affirmatively and summarize our contributions below.

**Circumventing Query-Composition Overhead with Per-Document Rényi Filters.** We propose a novel framework for multi-query differentially private RAG. Rather than composing a sequence of single-query DP-RAG executions, where the privacy budget grows with the number of queries, we leverage *individual Rényi filters* (Feldman & Zrnic, 2021). These filters bound privacy loss based on how many times each document is retrieved, yielding substantial savings when queries access largely

disjoint documents. To the best of our knowledge, this is the first application of privacy filters in the RAG setting. Our framework can incorporate any single-query private RAG algorithm.

**Two DP Multi-RAG Algorithms for Varying Test Query Dependencies.** We propose two differentially private RAG algorithms for the multi-query setting through threshold-based screening of relevant documents and their are tailored to the degree of relevance among test-time queries. MURAG (Algorithm 1) uses a fixed relevance threshold across all queries and is sufficient to work well for settings where queries are independent and do not share relevant private documents. MURAG-ADA (Algorithm 6) allocates a small portion of the privacy budget to release a query-specific relevance threshold, enabling more efficient use of the budget when queries are related and share overlapping relevant documents.

**Practical Multi-Query RAG with Non-Trivial Privacy Guarantees.** We evaluate our algorithms through extensive experiments on three LLMs (OPT-1.3B, Pythia-1.4B, and Mistral-7B). Our evaluation spans three types of datasets: standard RAG benchmarks (*Natural Questions*, *Trivia Questions*), a more challenging multi-hop QA dataset (MQuAKE) with correlated questions, and a privacy-sensitive application (ChatDoctor) consisting of patient–doctor QA pairs. Empirical results show that both of our methods can answer hundreds of queries within a total privacy budget of $\varepsilon \approx 10$ while maintaining reasonable utility, a trade-off no baseline method achieves. Furthermore, we demonstrate that our approaches with $\varepsilon = 10$ effectively defend against a state-of-the-art multi-query membership inference attack for RAG.

## 2 DIFFERENTIAL PRIVATE RETRIEVAL-AUGMENTED GENERATION

**Notation.** Let $\mathcal{V}$ denote a finite vocabulary, and let $x \in \mathcal{V}^*$ represent a prompt of arbitrary length. A document set of arbitrary size is denoted by $D = \{z_1, z_2, \ldots\}$, where each document $z_i \in \mathcal{V}^*$. For convenience, we define the document space as $\mathcal{Z} := 2^{\mathcal{V}^*}$. We use $\Delta$ to denote the symmetric difference between two sets. For sets $A$ and $B$, we define $A\Delta B := (A \setminus B) \cup (B \setminus A)$.

**Differential Privacy.** We denote the data space by $\mathcal{X}$. Two datasets $D, D' \in \mathcal{X}^*$ are said to be neighboring if they differ in at most one element. In this work, we study *document-level privacy* under the add/remove neighboring relation, where the data universe is $\mathcal{V}^*$ and two datasets are neighbors if they differ by exactly one document.

**Definition 1** (Differential Privacy (Dwork et al., 2006b)). *A randomized algorithm $\mathcal{M} : \mathcal{X}^* \to \Omega$ satisfies $(\varepsilon, \delta)$-differential privacy if, for all neighboring datasets $X, X' \in \mathcal{X}^*$ and all measurable subsets $O \subseteq \Omega$, $\Pr[\mathcal{M}(X) \in O] \leq e^\varepsilon \Pr[\mathcal{M}(X') \in O] + \delta$.*

**Definition 2** (Rényi Differential Privacy (Mironov, 2017)). *A randomized algorithm $\mathcal{M} : \mathcal{X}^* \to \Omega$ satisfies $(\alpha, \varepsilon)$-Rényi Differential Privacy (RDP) if, for all neighboring datasets $X, X' \in \mathcal{X}^*$, the Rényi divergence of order $\alpha > 1$ between $\mathcal{M}(X)$ and $\mathcal{M}(X')$ is at most $\varepsilon$, i.e. $D_\alpha(\mathcal{M}(X) \,\|\, \mathcal{M}(X')) \leq \varepsilon$.*

We may also consider *individual-level* RDP, where the Rényi divergence is evaluated on neighboring datasets that differ in a particular data point $z_i$. Let $\mathcal{S}(z_i, n)$ denote the set of dataset pairs $(S, \tilde{S})$ such that $|S|, |\tilde{S}| < n$ and $z_i \in S \triangle \tilde{S}$—i.e., exactly one of $S, \tilde{S}$ contains $z_i$.

**Definition 3** (Individual Rényi Differential Privacy). *A randomized algorithm $\mathcal{M} : \mathcal{X}^* \to \Omega$ satisfies $(\alpha, \varepsilon)$-individual RDP at point $z_i$ if, for all $(X, X') \in \mathcal{S}(z_i, n)$, $D_\alpha(\mathcal{M}(X) \,\|\, \mathcal{M}(X')) \leq \varepsilon$*

A *privacy filter* is a stopping rule that tracks cumulative privacy loss and halts execution once the privacy budget is exceeded, thereby ensuring that the designed privacy guarantees are never violated. For completeness, we briefly introduce individual RDP filters; for a rigorous treatment, we refer readers to Feldman & Zrnic (2021).

**Definition 4** ((Individual) Rényi Differential Privacy Filters (Feldman & Zrnic, 2021)). *A random variable $\mathcal{F}_{\alpha,B} : \Omega^* \to \{\text{CONT}, \text{HALT}\}$ is a privacy filter for $(\alpha, B)$-RDP if it halts the execution of an algorithm before its accumulated (individual) privacy loss, measured in $\alpha$-Rényi divergence, exceeds $B$.*

**Problem Setting.** We study retrieval-augmented generation (RAG) with a sensitive external document collection. A decoder-only LLM with greedy decoding is modeled as a function

LLM $: \mathcal{V}^* \times \mathcal{Z} \to \mathcal{V}$. Given a user prompt $x \in \mathcal{V}^*$, the system retrieves a subset of documents $D_x = R_k(x, D)$ from a private external corpus $D \in \mathcal{Z}$, where the retrieval function $R_k : \mathcal{V}^* \times \mathcal{Z} \to \mathcal{Z}$ returns the $k$ most relevant documents. The corpus $D$ contains sensitive documents, each potentially corresponding to private user information.

We adopt a threat model in which the adversary has no direct access to the corpus $D$ but may issue arbitrary prompts $x$ to the RAG system. The underlying LLM is assumed to be public and independent of $D$. Our objective is to design a differentially private RAG mechanism that, given a set of queries $\{q_1, \ldots, q_T\}$, the sensitive corpus $D$, a public LLM, and a total privacy budget $\varepsilon$, generates high-utility responses while guaranteeing $\varepsilon$-differential privacy with respect to corpus $D$.

## 3 METHODOLOGY

### 3.1 TECHNICAL OVERVIEW

**Improved Privacy Accounting via Per-Document Privacy Filters.** In retrieval-augmented generation (RAG), each query interacts with only a small, query-specific subset of the corpus $D$. This sparsity implies that most documents are accessed only rarely. We leverage this by introducing a per-document privacy filter that monitors cumulative privacy loss and blocks further retrieval once a document's budget is exhausted. Because privacy cost is incurred only upon retrieval, this accounting scheme naturally scales with the frequency of document access rather than the total number of queries.

**Screening Relevant Documents via Relevance Thresholding.** If RAG were applied directly to the entire corpus, every document would be touched by each query, and per-document privacy filters would provide no benefit. To prevent this, MURAG employs a global relevance threshold $\tau$[1]: only documents whose scores exceed $\tau$ are retrieved and incur privacy cost. A document is excluded from all future retrievals once its privacy budget is exhausted. Since $\tau$ is fixed in advance and independent of the data, introducing this threshold does not consume additional privacy budget.

**Handling Correlated Queries via Adaptive Thresholding.** When queries are *correlated*, meaning their sets of relevant documents substantially overlap, a fixed relevance threshold $\tau$ can lead to inefficiencies. Specifically, since the relevance score distribution may shift across queries, a uniform threshold can cause some queries to retrieve more documents than necessary, prematurely exhausting the budgets of relevant documents and limiting their availability for later queries. To mitigate this, we propose MURAG-ADA, which privately selects a query-specific threshold $\tau_t$ tailored to the relevance distribution of each query. By combining per-document privacy accounting with the private release of cumulative statistics, MURAG-ADA restricts retrieval to the most relevant documents, thereby reducing unnecessary budget consumption and preserving utility across correlated queries.

**Single-Query DP RAG after Screening.** After thresholding, per-document privacy filters ensure that each retrieved document incurs loss only when used and is removed once its budget is exhausted. The resulting set is then passed to a single-query DP-RAG algorithm to generate the response. As shown in Algorithms 1 and 6, our multi-query framework is modular, supporting any private single-query RAG method. In this work, we instantiate it with a pure-DP variant of the algorithm from Koga et al. (2024) (Algorithm 4).

### 3.2 DP-RAG WITH A FIXED THRESHOLD

In MURAG, we impose a fixed relevance threshold $\tau$ to screen documents before retrieval. The threshold can either be publicly specified or privately estimated using a small portion of the privacy budget. The complete procedure is summarized in Algorithm 1 and the privacy guarantee is given in Theorem 1. At a high level, the algorithm maintains a per-document privacy budget that is decremented whenever the document is retrieved. For each query, it first updates the active set of documents and then filters out most documents with scores below $\tau$. Among the remaining documents, the top-$k$ are selected by relevance, and a differentially private single-query RAG procedure is invoked to generate the response.

---

[1]Intuitively, the threshold $\tau$ can be viewed as a chosen percentile of the relevance score distribution for a given query, ensuring that only the top-ranked documents contribute to privacy cost.

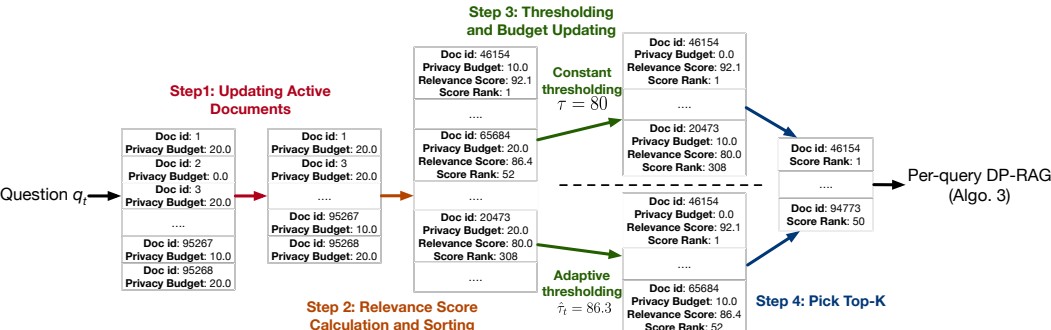

Figure A: General workflow of our multi-query DP-RAG algorithms MURAG and MURAG-Ada (all numerical values are provided for illustrative purposes only). For each incoming query $q_t$, *Step 1*: update the active document set by removing documents whose individual privacy budgets are exhausted or fall below a prescribed threshold (eg. document 2 is removed because its privacy budget has been exhausted); *Step 2*: compute and sort relevance scores for all active documents; *Step 3*: apply either a fixed relevance threshold $\tau$ (MURAG, top branch) or an adaptive threshold $\hat{\tau}_t$ (MURAG-Ada, bottom branch) to determine which documents enter the screened set and update their remaining individual privacy budget (eg. the privacy budget of document 20473 is reduced by 10); *Step 4*: run a top-K selection procedure over the screened documents to refine the retrieved set further. Finally, the selected top-K documents are passed to the per-query DP-RAG mechanism to generate the final answer.

Since whether a document exceeds the constant threshold $\tau$ depends only on its own score and not on the scores of other documents, the use of *(Individual) Rényi Differential Privacy Filters* is valid. Consequently, for each query, privacy loss is charged only to the small subset of documents that pass the threshold, using a per-query budget $\varepsilon_q$, rather than to the entire corpus.

---

**Algorithm 1:** MURAG: Differentially Private **Mu**lti-Query **R**etrieval-**A**ugmented **G**eneration

---

**Input:** Private dataset $D$, sequence of queries $\{q_1, \ldots, q_T\}$, per-query DP budget $\varepsilon_q$, #retrieved documents $k$, maximum retrievals per document $M$, relevance threshold $\tau$

**Set:** Initialize individual budget for each document $z \in D$: $\mathcal{E}(z) = M \cdot \varepsilon_q$

1 **for** $t = 1, \ldots, T$ **do**
2     $A_t = \{z \in D \mid \mathcal{E}(z) \geq \varepsilon_q\}$             ▷ Update active document set
3     $D_{q_t} = \{z \in A_t \mid r(z, q_t) > \tau\}$           ▷ Filter relevant documents
4     **for** $z \in D_{q_t}$ **do**
5        $\mathcal{E}(z) \leftarrow \mathcal{E}(z) - \varepsilon_q$         ▷ Update budget for retrieved documents
6     $D_{q_t}^k = \text{TOP-K}(D_{q_t}, k, r(\cdot, q_t))$      ▷ Select top-$k$ relevant documents
7     $a_t = \text{DP-RAG}(x, D_{q_t}^k, \text{LLM}, \varepsilon_q)$      ▷ Generate DP response via Algo. 4
8 **return** $(a_1, \ldots, a_T)$

---

**Theorem 1** (Privacy Guarantee of Algorithm 1). MURAG *satisfies $\varepsilon$-differential privacy provided that the initial privacy budget assigned to each document $z \in D$ is at most $\varepsilon$.*

### 3.3 DP-RAG WITH ADAPTIVE THRESHOLD

The score distribution can vary substantially across different questions, making a single global threshold ineffective. To guarantee the performance of single-query DP-RAG, the threshold must be set low enough to retrieve sufficient documents for all queries. However, this often results in many unnecessary documents being retrieved: although single-query DP-RAG uses at most $K$ documents, any additional documents above $K$ still incur privacy loss, wasting budget on unused data. This inefficiency can significantly degrade performance when those documents are needed by later queries. To overcome this limitation, we propose MURAG-ADA, which privately releases a query-specific threshold $\tau_t$ adapted to the relevance distribution of each query.

The adaptive procedure works by discretizing the relevance scores into bins and then releasing noisy prefix sums until the cumulative count of retrieved documents exceeds $K$. This mechanism tailors the cutoff of documents to each query, reducing unnecessary budget consumption on irrelevant

documents and preserving utility across multiple queries. We will see in the experimental section that this approach especially yields clear utility gains on datasets with high correlated queries. The full procedure is summarized in Algorithm 6. (Appendix C.4)

Notice that in Algorithm 6, we use $k$ as a stopping criterion instead of releasing differentially private top-$k$ relevance scores. This is because releasing a noisy top-$k$ score for each query would make the privacy budget grow linearly with the number of queries and incur loss on all documents, thereby breaking the per-document privacy filter. By contrast, our prefix-sum approach (Step 1 of Algorithm 6) incurs privacy loss only on the documents that appear in the released prefix sums, while all other documents remain untouched. This concentrates the privacy cost of this step still on a small subset, yielding tighter accounting and more efficient budget use across multiple queries.

---

**Algorithm 2:** MuRAG-ADA: DP **Mu**lti-Query **RAG** with **Ada**ptive Threshold

---

**Input:** Private dataset $D$, sequence of queries $\{q_1, \ldots, q_T\}$, per-query budget $\varepsilon_q$, number of retrieved documents $k$, maximum retrievals per document $M$

**Set:** Initialize budget for each $z \in D$: $\mathcal{E}(z) \leftarrow M \cdot \varepsilon_q$. Split budget: $\varepsilon_q = \varepsilon_{\text{thr}} + \varepsilon_{\text{RAG}}$.

**Require:** Discretization of similarity scores into bins $[a_i, a_{i+1})_{i=1}^B$

1  **for** $t = 1, \ldots, T$ **do**
     /* Step 1:  Adaptive thresholding via noisy prefix sums */
2    $\tilde{s} \leftarrow 0, A_t \leftarrow \varnothing$
3    **for** $i = 1, \ldots, B$ **do**
4        $A_t^{(i)} = \{z \in D \mid r(z, q_t) \in [a_i, b_i], \mathcal{E}(z) \geq \varepsilon_{\text{thr}}\}$
5        $\tilde{s} \leftarrow \tilde{s} + |A_t^{(i)}| + \text{Lap}(1/\varepsilon_{\text{thr}})$
6        $A_t \leftarrow A_t \cup A_t^{(i)}$
7        **for** $z \in A_t^{(i)}$ **do**
8            $\mathcal{E}(z) \leftarrow \mathcal{E}(z) - \varepsilon_{\text{thr}}$
9        **if** $\tilde{s} \geq k$ **then**
10           $\tau_t = a_i$; break                            ▷ Release threshold

     /* Step 2:  DP-RAG on adaptively selected active set */
11   $A_t' = \{z \in A_t \mid \mathcal{E}(z) \geq \varepsilon_{\text{RAG}}\}$
12   $D_{q_t} = \text{Top-K}(A_t', k, r(\cdot, q_t))$
13   $a_t = \text{DP-RAG}(x, D_{q_t}, \text{LLM}, \varepsilon_{\text{RAG}}; \tau_t)$        ▷ single-query RAG, Algo. 4
14   **for** $z \in A_t'$ **do**
15       $\mathcal{E}(z) \leftarrow \mathcal{E}(z) - \varepsilon_{\text{RAG}}$

16 **return** $(a_1, \ldots, a_T)$

---

**Theorem 2** (Privacy Guarantee of Algorithm 6). MuRAG-ADA *satisfies $\varepsilon$-differential privacy provided that the initial privacy budget allocated to each document $z \in D$ is at most $\varepsilon$.*

## 4 EXPERIMENT

### 4.1 DATASET

**Datasets set-up.** We first evaluate our methods on **two independent question sets**: *Natural Questions* and *Trivia Questions*. These are standard benchmarks for evaluating RAG systems and have been used in prior work on per-query DP for RAG (Koga et al., 2024). Following their setup, we randomly subsample 100 questions from each dataset to reduce computational overhead. Importantly, the questions are independent of one another, and each requires a disjoint set of relevant documents from the external database. To quantify document reuse, we examine how frequently each document appears in the top-$K$ retrieved results ($K = 50$) across questions. As shown in Figure 1, in both *Natural Questions* and *Trivia Questions*, most documents are retrieved for only one or two queries. *Thus, we expect* MuRAG *to perform sufficiently well on these two datasets.*

Second, we consider a **correlated question set**, *MQuAKE* (Zhong et al.). This dataset contains sequences of semantically related single-hop questions that together form multi-hop reasoning chains.

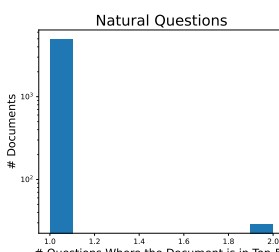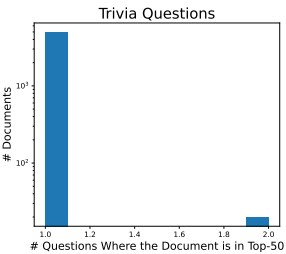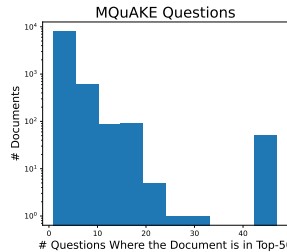

Figure 1: Histogram of document reuse across questions. Each bar shows how many questions a document appears in among the top-$K$ retrieved results ($K = 50$). The x-axis indicates the number of questions per document, and the y-axis shows the count of such documents.

We select 100 such sequences, yielding 400 individual questions for evaluation. Since questions in the same sequence share entities (subjects or objects), their relevant documents substantially overlap. As shown in Figure 1, many documents appear across multiple questions. *We therefore expect* MURAG-ADA *to have an advantage over* MURAG.

Finally, we evaluate on *ChatDoctor* (Li et al., 2023), a **privacy-sensitive application of RAG** in the healthcare domain. This dataset consists of QA interactions between patients and doctors. We sample 100 patient questions as our test set. *This evaluation tests the effectiveness of our methods in a real-world sensitive setting and their robustness against privacy attacks.*

**External datasets reflecting both standard and privacy-sensitive settings.** For Natural Questions, Trivia Questions, and MQuAKE Questions, we use Wikipedia of $\sim 20M$ documents as the external knowledge source following the standard RAG setup (Chen et al., 2017; Lewis et al., 2020). For ChatDoctor Questions, the external dataset consists of the remaining $\sim 200K$ QA pairs from the original ChatDoctor dataset, excluding the 100 patient questions used for testing. This setup reflects a realistic privacy-sensitive application, where the external corpus contains private information.

**QA evaluation metric.** For Natural Questions, Trivia Questions and MQuAKE Questions, the datasets provide a list of all acceptable correct answers for each question. Following the evaluation protocol of Koga et al. (2024), we use the *Match Accuracy* metric: a prediction is scored as 1 if it contains any correct answer, and 0 otherwise. For Chatdoctor Questions, we adopt the evaluation metric from the original dataset paper, using the F1 score of BERTScore (Zhang et al., 2020) to measure semantic similarity between the predicted response and the ground-truth answer.

## 4.2 MODEL AND METHOD SET-UP

**Model set-up.** Our RAG pipeline integrates three pre-trained LLMs: OPT-1.3B (Zhang et al., 2022), Pythia-1.4B (Biderman et al., 2023), and Mistral-7B (Jiang et al., 2023). For document retrieval, we use the Dense Passage Retriever (DPR) (Karpukhin et al., 2020) to compute dense query-document relevance scores.

**Baseline methods.** We compare our two proposed methods with four baselines. The first is **NAIVE-MULTI-RAG** (Algorithm 5), which applies the per-question DP RAG method, DPSparse-VoteRAG, independently to each query and uses the standard sequential composition theorem (Dwork et al., 2006a) to compute the overall privacy guarantee. The second baseline privatizes the external dataset of RAG under differential privacy (DP) and then uses the resulting synthetic dataset as the knowledge source for evaluation. In this setup, the answers are guaranteed to satisfy DP since they are derived from a privatized dataset. We adopt **Private Evolution** (PE; Xie et al. (2024)), a state-of-the-art DP synthetic text generation method that also aligns with the query-access setting of RAG. Specifically, PE first queries an LLM to produce an initial dataset within the same domain as the private corpus, and then refines its distribution under DP to better approximate that of the private dataset. To ensure consistency, for each pretrained LLM used in RAG, we use the same model as the query API in PE. The other two are non-private baselines: **Non-RAG**, which generates answers using the pretrained LLM without retrieval, and **Non-Private-RAG**, which performs retrieval-augmented generation without any privacy mechanism. We describe implementation details in Appendix E.

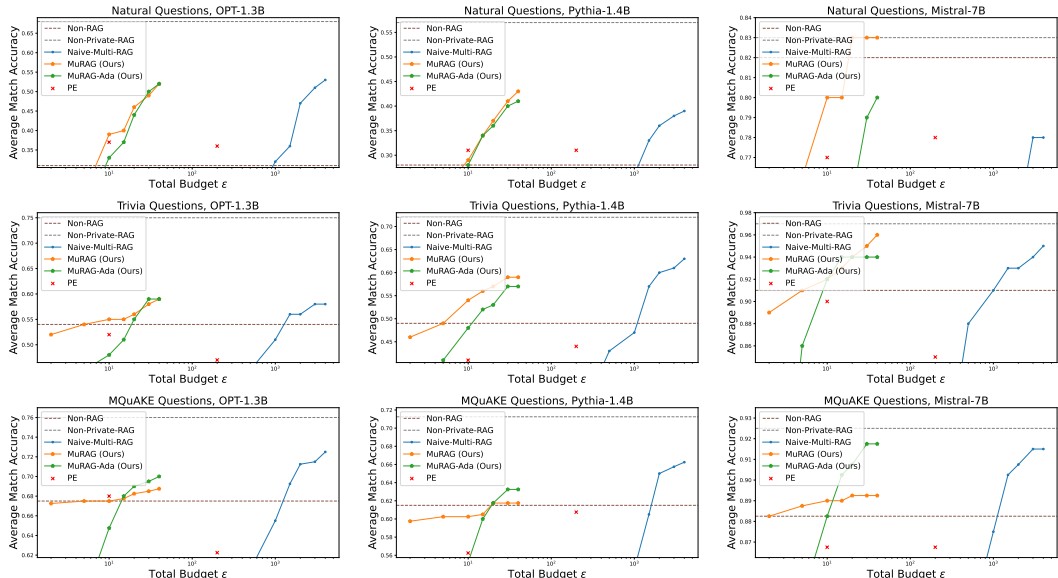

Figure 2: Privacy-Utility tradeoffs of our two proposed methods (MURAG and MURAG-ADA) compared to baselines across three pretrained LLMs and two categories of question sets.

**Privacy budget setup for DP algorithms.** Following the setup in Koga et al. (2024), we vary the per-query RAG privacy budget $\varepsilon_q \in \{2, 5, 10, 15, 20, 30, 40\}$ to explore the privacy-utility trade-off. For NAIVE-MULTI-RAG, the total privacy budget is $T \cdot \varepsilon_q$, where $T$ is the number of questions. For MURAG and MURAG-ADA, the total budget is $M \cdot \varepsilon_q$, where $M$ is the number of retrieved documents with nonzero privacy loss[2]. In our main results, we conservatively set $M = 1$ for a realistic privacy region. Moreover, $\varepsilon_{\text{thr}}$ is fixed as $1.0$. For the baseline PE, we test with $\varepsilon \in \{10, 200\}$.

**Membership inference attack in RAG.** To assess the effectiveness of our privacy-preserving methods, we evaluate them against the membership inference attack (MIA). The objective of MIA is as follows: given a candidate document $x$ and a model system $R(\cdot; D)$ trained on a private dataset $D$, the adversary aims to determine whether $x \in D$ by computing a membership score $s(x, R(\cdot; D))$. Without loss of generality, we assume higher scores indicate higher membership likelihood. Applying the attack to an in-distribution set $D_{\text{in}} \subset D$ and an out-of-distribution set $D_{\text{out}}$ (with no overlap with $D$) allows us to derive the TPR–FPR curve and compute the AUC, which serves as the evaluation metric for attack success.

We focus on scenarios where the adversary can issue multiple queries to the system, as this setting substantially amplifies the attack strength. To model this, we adopt the *Interrogation Attack (IA)* (Naseh et al., 2025), a state-of-the-art MIA specifically designed to exploit multi-query access in RAG systems. For each document $x$, IA generates $m = 30$ tailored questions together with their corresponding answers implied by $x$. Then each question is concatenated with the necessary context to ensure the target document can be retrieved, and the query is then submitted to the RAG system. The membership score is defined as the accuracy of the RAG system across these $m$ questions, where higher accuracy implies a greater likelihood that the document is present in the external dataset and is being retrieved to answer the queries. Additional implementation details, including the question generation process, are provided in Appendix E.

### 4.3 MAIN RESULTS

**Results on two standard RAG benchmarks (independent question sets).** Figure 2 shows the performance of our two proposed methods compared with three baselines across three pretrained LLMs on *Natural Questions* and *Trivia Questions*. Both of our methods outperform the Non-RAG baseline in most cases under a total privacy budget of $\varepsilon = 10$.

---

[2]To enable a meaningful comparison, we convert our privacy guarantee, originally expressed in $(\infty, \varepsilon)$-RDP, into an equivalent $\varepsilon$-DP guarantee (Mironov, 2017).

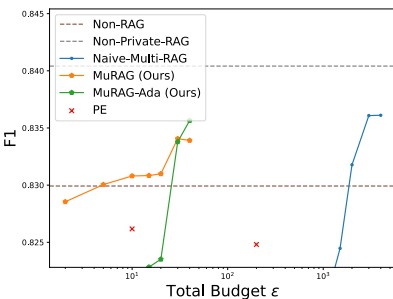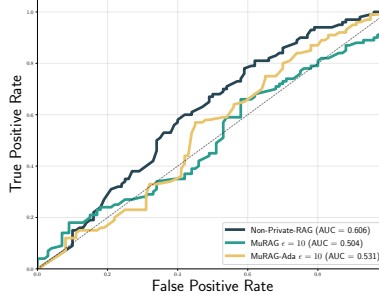

Figure 3: **Left:** Privacy-utility tradeoffs of our two methods and baselines. **Right:** TPR-FPR curves of IA (Membership Inference Attack with multiple queries). Both experiments are conducted with Mistral-7B and ChatDoctor datasets.

In contrast, the baseline NAIVE-MULTI-RAG requires an impractically large budget, exceeding $\varepsilon = 10^3$, to achieve comparable utility. This highlights that our approaches make differential privacy practical in the multi-query RAG setting by leveraging more tailored compositions, enabling strong utility within a realistic privacy budget. The PE baseline performs even worse than Non-RAG at $\varepsilon = 200$ for many settings, which we attribute to objective misalignment: PE optimizes for distributional similarity (e.g., measured by Fréchet Inception Distance (FID; (Heusel et al., 2017))) rather than preserving factual content. Indeed, we find PE achieves a better FID score at $\varepsilon = 200$ but yields lower task performance than at $\varepsilon = 10$ on the setting of Trivia Questions and OPT-1.3B, further supporting this explanation.[3]

Lastly, on these two datasets, MURAG outperforms MURAG-ADA, which aligns with our expectations. Since the questions are independent, adaptive thresholding provides little benefit and additionally consumes extra privacy budget.

**Results on multi-hop questions (correlated question set).** Figure 2 shows the performance of our two proposed methods compared with three baselines across three pretrained LLMs on *MQuAKE Questions*. Overall, the relative trends between our methods and the baselines are consistent with the independent question setting. However, a key difference emerges in the comparison between our two approaches: MURAG-ADA performs significantly better than MURAG. This result is aligned with our intuition, as adaptive thresholding is particularly advantageous when questions are correlated and share overlapping relevant documents.

**Results on privacy-sensitive application.** The left panel of Figure 3 illustrates that MURAG and MURAG-ADA achieve a better privacy–utility trade-off than both prior two DP-based approaches on Mistral-7B with the ChatDoctor dataset. At the same privacy budget $\varepsilon = 10$, both of our methods not only outperform the DP baselines (Naive multi-RAG and Private Evolution), which suffer from severe utility degradation, but even surpass the Non-RAG baseline in this practical, privacy-sensitive setting. We further evaluate robustness against the Interrogation Attack (IA) and a data-extraction attack on ChatDoctor, comparing three RAG systems: Non-Private-RAG, MURAG ($\varepsilon = 10$), and MURAG-ADA ($\varepsilon = 10$). For IA, the right panel of Figure 3 shows that, without protection, IA attains a non-trivial AUC of $\approx 0.6$, whereas both of our methods drive the AUC down to $\approx 0.5$, effectively neutralizing the attack. For the data extraction attack, Figure 7 (Appendix F.5) reports the similarity-score distributions. For BERTScore, the non-private RAG exhibits a higher mode ($\approx 0.864$ vs. $\approx 0.85$ for the private variants) and much heavier high-similarity tails (scores $> 0.884$), while both MURAG and MURAG-ADA remain below this range. For ROUGE-L, the two private RAG methods have very light upper tails, in contrast to the non-private RAG, which has $12\%$ of points with ROUGE-L $> 0.3$, indicating greater overlap with reference documents and thus higher privacy risk. Taken together, these results show that MURAG and MURAG-ADA deliver strong utility and provide practical privacy protection at $\varepsilon = 10$ in a real-world sensitive application.

---

[3]We confirm that the FID score improves from $\varepsilon = 10$ to $\varepsilon = 200$ (0.066 to 0.036; lower is better) on the setting of Trivia Questions and OPT-1.3B, yet RAG utility drops, underscoring the mismatch between FID and factual fidelity required for RAG.

Table 1: Precision of retrieved documents under different thresholding approaches, measured as the percentage of truly top-50 relevant documents among the retrieved.

| | Independent Question Set | | Correlated Question Set |
| --- | --- | --- | --- |
| | Natural Questions | Trivia Questions | MQuAKE Questions |
| Constant Thresholding (in MuRAG) | 78.8% | 72.2% | 17.6% |
| Adaptive Thresholding (in MuRAG-ADA) | 92.6% | 94.6% | 40.7% |
| Adaptive Thresholding (Non-private top-K-release) | 99.4% | 99.6% | 43.5% |

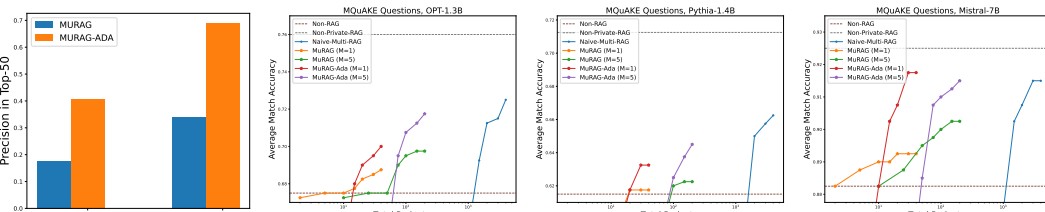

Figure 4: Comparison of $M = 1$ and $M = 5$ in the individual privacy accounting framework. The left plot shows the retrieval precisions of two methods with $M = 1, 5$. Right three plots show the trade-off between the QA performance and the $\varepsilon_{\text{total}}$ in DP.

**Takeaway.** Across all evaluations, our methods consistently outperform baseline approaches under practical privacy budgets. On independent question sets, MuRAG achieves strong performance as expected, while on correlated multi-hop questions, MuRAG-ADA shows clear advantages due to its adaptive thresholding. Finally, in the privacy-sensitive ChatDoctor application, both methods not only improve privacy-utility trade-off over baselines but also effectively mitigate both state-of-the-art membership inference attacks and data extraction attacks. Together, these results demonstrate that our approaches make differentially private RAG both practical and robust across diverse settings.

### 4.4 FURTHER ANALYSIS OF MuRAG AND MuRAG-ADA

**Comparison between thresholding approaches in our two methods.** The two methods have different performance as discussed above, and the difference is between the constant thresholding and the DP-released adaptive thresholding. To quantify this effect, Table 1 reports the precision under both *constant thresholds* (in MuRAG) and *adaptive thresholds* (in MuRAG-ADA), where we measure the percentage of truly top-50 documents among the retrieved documents for each question and calculate the average over questions as the precision. We observe that precision under MuRAG is particularly low for the correlated question set *MQuAKE Questions*, whereas MuRAG-ADA significantly improves retrieval precision on these datasets through its adaptive thresholds. This improvement in retrieval quality directly contributes to the superior performance of MuRAG-ADA in the setting of *correlated question set*.

**Effect of different $M$ in the individual privacy accounting framework.** Both of our proposed methods include a hyperparameter $M$, which controls the maximum number of queries for which an individual document's privacy budget can be consumed. In our main results (Figure 2), we set $M = 1$ to ensure strict per-document privacy usage. However, this setting may limit utility: once a document is used for one query, it becomes unavailable for future queries, even if it would have been highly relevant. To better understand the impact of $M$, we evaluate our two methods with a larger value of $M = 5$. The left plot in Figure 4 shows a substantial increase in Top-50 retrieval precision when using $M = 5$, indicating better access to relevant documents. This improvement translates into higher end-to-end RAG utility, as shown in the three plots on the right. However, increasing $M$ also leads to a higher total privacy cost ($\varepsilon_{\text{total}} = M \cdot \varepsilon_q$).

## 5 RELATED WORK

Recent studies identify two main privacy risks in retrieval-augmented generation (RAG) systems. The first is membership inference attacks (MIA) (Shokri et al., 2017), which test whether a specific document is in the private external dataset, often via adversarial prompts (Naseh et al., 2025; Liu

et al., 2025; Anderson et al., 2024) or scoring mechanisms (Li et al., 2025). The second is data reconstruction attacks, which aim to recover document content using adversarial prompts (Zhang et al., 2025; Zeng et al., 2024a; Jiang et al., 2024) or poisoning triggers (Peng et al., 2024). Together, these works highlight the growing need for principled privacy-preserving algorithms for RAG.

Several DP-based defenses have been proposed. Koga et al. (2024) introduced a single-query DP-RAG system, and others (Yao & Li; Grislain, 2025) studied DP release of document identifiers. However, none of these methods address the realistic multi-query setting. In addition to DP based methods, empirical defenses have also been explored, including paraphrasing retrieved documents (Yao & Li) and dataset privatization (Zeng et al., 2024b), but these lack formal privacy guarantees and remain vulnerable to strong adversarial attacks. A complementary line of work considers protecting user queries in cloud-hosted RAG (Cheng et al., 2024), which addresses a different threat model than ours.

For additional related work on the use of differential privacy in large language models and the line of individual privacy accounting, we refer readers to Appendix B.

## 6 DISCUSSION

**Why Privacy Filter rather than Amplification by Subsampling?**    As surveyed in Section B.1, privacy amplification by subsampling (Balle et al., 2018; Wang et al., 2019; Zhu & Wang, 2019) is widely used in DP LLM applications, such as DP prompt tuning and DP in-context learning, to enhance generation quality. However, this technique is not well-suited for DP RAG:

- In prompt tuning, the goal is to learn a single task-specific prompt that can generalize to all future queries. In DP in-context learning, a small number of example inputs are selected under DP constraints and reused across queries. In contrast, RAG does not allow for such "unified" prompts or examples: each test-time query requires retrieving and using query-specific documents, which must be handled privately, which makes individual privacy filter a more suitable choice.

- Moreover, in prompt tuning and in-context learning, all data points in the private dataset can meaningfully contribute to the learned prompt or selected example set. This property enables the use of subsampling-based amplification techniques in algorithm design. In RAG, however, only a sparse subset of documents in the large external corpus are relevant to any given query—most documents provide no utility.

These two key differences, the lack of reusable prompts and the sparsity of useful data, motivate the development of our new DP RAG algorithms using Rènyi filter rather than amplification by sampling.

**Leveraging Historical QA.**    As shown in Table 1 and Figure 1, when the relevant documents for different questions exhibit significant overlap, the quality of answers to later questions degrades. This occurs because the documents required to answer the queries may exhaust their privacy budgets and are subsequently filtered out from the active set passed to the RAG algorithm. In the extreme case where a user repeatedly submits the same query, only the first response may retain high quality, while subsequent answers degrade due to the unavailability of relevant documents.

A potential remedy is to reuse historical answers as auxiliary documents in future queries. This can be done without incurring any additional privacy cost, owing to the post-processing property of differential privacy.

## 7 CONCLUSION

We proposed the first differentially private (DP) framework for retrieval-augmented generation (RAG) that supports answering multiple queries while protecting a sensitive external dataset. We introduced two algorithms: MURAG and MURAG-ADA differ in how they select documents for each query under DP guarantees, which have their advantage for different types of question set. Through comprehensive experiments on various question datasets and three LLMs, we demonstrated that our methods achieve the utility that outperforms a Non-RAG baseline for answering 100 questions under a realistic budget of $\varepsilon = 10$. We also showed that MURAG-ADA performs particularly well on correlated question sets. We hope our contributions provide a foundation for more practical and principled privacy-preserving RAG systems.

REPRODUCIBILITY STATEMENT

We provide the detailed pseudocode in Algorithm 1 and Algorithm 6. We provide the detailed implementation information in Appendix E.

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

## A  THE USE OF LLMs

We employ large language models (LLMs) primarily to improve the grammar and clarity of our writing. All research ideas, directions, and decisions, however, are independently conceived and carried out by the authors.

## B  EXTENDED RELATED WORK

### B.1  DIFFERENTIAL PRIVACY IN LARGE LANGUAGE MODELS

Beyond our focus on DP for RAG, differential privacy has also been explored in a variety of LLM settings, including pre-training and fine-tuning (Charles et al., 2024; Yu et al., 2021; Li et al., 2021), prompt tuning (Duan et al., 2023; Hong et al., 2024), and in-context learning (Tang et al., 2024; Wu et al., 2024). These tasks differ structurally and thus require different DP mechanisms. In pre-training and fine-tuning, the challenge lies in optimizing model parameters while maintaining stability under DP noise, whereas in RAG, the emphasis is on protecting privacy during inference-time retrieval and generation. Closer to our setting are DP methods for prompt tuning and in-context learning. Still, the structural differences between these tasks and RAG lead to distinct algorithmic requirements (see Section 6 for discussion). Another line of research investigates differentially private synthetic test generation under varying levels of model access. Vinod et al. (2025); Amin et al. (2025; 2024) focus on next-token prediction with logits access, while Xie et al. (2024) studies the API-access setting, which we also include in our comparisons.

### B.2  INDIVIDUAL PRIVACY ACCOUNTING AND PRIVACY FILTERS

Individual privacy accounting tracks the privacy loss of a single data point, often yielding tighter bounds than worst-case analyses over all neighboring datasets (Dwork et al., 2006b). This perspective was introduced by Feldman & Zrnic (2021) in the context of Rényi Differential Privacy and later extended to Gaussian Differential Privacy by Koskela et al. (2022). See Feldman & Zrnic (2021, Section 1.2) for a detailed overview. Within this framework, privacy filters provide a general mechanism for adaptively enforcing privacy constraints by halting an algorithm once the cumulative privacy loss reaches a budget. Individual privacy filters (Feldman & Zrnic, 2021; Koskela et al., 2022) refine this idea by operating at the granularity of single data points, excluding them from further computation once their budgets are exhausted. For additional developments and extensions, see Rogers et al. (2016); Feldman & Zrnic (2021); Koskela et al. (2022); Smith & Thakurta (2022); Whitehouse et al. (2023).

## C  ALGORITHMS

In this appendix, we provide additional algorithms that were excluded from the main text for space considerations.

### C.1  TOP-K DOCUMENT SELECTION

Algorithm 3 selects the top-$K$ documents from dataset $D$ according to the score function $r$. If $|D| < K$, the output is padded with empty strings so that it always has exactly $K$ elements, which is required for the privacy accounting (see Lemma 2).

---

**Algorithm 3:** TOP-K$(D, K, r)$

**Input:** dataset $D$, sample size $K$, score function $r$
1 **if** $|D| \geq K$ **then**
2     $D^K \leftarrow$ top-$K$ documents from $D$ ranked by $r$        ▷ assume no ties
3 **else**
4     $D^K \leftarrow D \cup \{\texttt{""}\}^{K-|D|}$        ▷ pad with empty strings to size $K$
5 **return** $D^K$

---

## C.2 DP-RAG for single question answering

Algorithm 4 is a variant of Koga et al. (2024, Algorithm 2), where the LimitedDomain mechanism (Durfee & Rogers, 2019) is replaced by the exponential mechanism in the private token generation step. This modification provides a stronger pure-DP guarantee and simplifies the privacy composition analysis.

---

**Algorithm 4:** DP-RAG$(x, D, \text{LLM}, \varepsilon)$

---

**Input:** Prompt $x$; external data source $D$; LLM(prompt, doc | history); total budget $\varepsilon$;
**Set:** Per-token privacy budget $\varepsilon_0$
**Require:** maximum length of output tokens $T_{\max}$; number of voters $m$; retrievals per voter $k$;
    document retriever $R$(prompt, doc set, #retrieved docs); threshold for voting $\theta$

1   $\varepsilon_{\text{Expo}} \leftarrow \varepsilon_0/2, \varepsilon_{\text{Lap}} \leftarrow \varepsilon_0/2$    ▷ split privacy budget for per token generation

2   $c \leftarrow \lfloor \varepsilon/\varepsilon_{\text{RAG}} \rfloor, \hat{\theta} \leftarrow \theta + \text{Lap}(2/\varepsilon_{\text{Lap}})$

3   $D_x \leftarrow R(x, D; mk)$        ▷ retrieve $mk$ documents

4   $\mathcal{D}_x \leftarrow \{D_x^1, \ldots, D_x^m\}$    ▷ Partition $D_x$ into $m$ subsets uniformly random

5   **for** $t \leftarrow 1$ **to** $T_{\max}$ **do**

6     $y_t^{\text{non-RAG}} \leftarrow \text{LLM}(x, \text{""} \mid y_{<t})$

7     **for** $i \leftarrow 1$ **to** $m$ **do**

8       $y_t^{(i)} \leftarrow \text{LLM}(x, D_x^i \mid y_{<t})$

9     $\text{Hist}_t \leftarrow \text{Hist}(y_t^{(1)}, \ldots, y_t^{(m)})$      ▷ $\text{Hist}_t \in \mathbb{N}^{|\mathcal{V}|}$

10    $\text{Count}_t \leftarrow \text{Hist}_t[\text{index} = y_t^{\text{non-RAG}}]$

11    **if** $\text{Count}_t + \text{Lap}(4/\varepsilon_{\text{Lap}}) \leq \hat{\theta}$ **then**

12      $y_t \leftarrow \text{expoMech}(\text{Hist}_t; \varepsilon_{\text{Expo}})$

13      $c \leftarrow c - 1$

14    **else**

15      $y_t \leftarrow y_t^{\text{non-RAG}}$

16    **if** $y_t = \langle \text{EOS} \rangle$ **or** $c = 0$ **then**

17      **return** $(y_1, \ldots, y_t)$

18 **return** $(y_1, \ldots, y_{T_{\max}})$

---

## C.3 Naive algorithm for DP Multi-Query RAG

Algorithm 5 serves as a baseline approach to multi-query DP-RAG, obtained by composing a single-query DP-RAG mechanism sequentially for $T$ rounds.

---

**Algorithm 5:** NAIVE-MULTI-RAG

---

**Input:** Private external dataset $D$, query sequence $\{q_1, q_2, \ldots, q_T\}$, total privacy budget $\varepsilon$,
   per-query budget $\varepsilon_q$
**Require:** $\varepsilon \geq T \cdot \varepsilon_q$

1 **for** $t = 1, \ldots, T$ **do**

2    $a_t \leftarrow \text{DP-RAG}(q_t, D, \text{LLM}, \varepsilon_q)$      ▷ Apply Algorithm 4

3 **return** $(a_1, a_2, \ldots, a_T)$

---

## C.4 DP-RAG WITH ADAPTIVE THRESHOLD

In this section, we present our DP multi-query RAG algorithm with adaptive thresholding. A detailed discussion is provided in Section 3.3.

---

**Algorithm 6:** MURAG-ADA: DP **Mu**lti-Query **RAG** with **Ada**ptive Threshold

---

**Input:** Private dataset $D$, sequence of queries $\{q_1, \ldots, q_T\}$, per-query budget $\varepsilon_q$, number of retrieved documents $k$, maximum retrievals per document $M$

**Set:** Initialize budget for each $z \in D$: $\mathcal{E}(z) \leftarrow M \cdot \varepsilon_q$. Split budget: $\varepsilon_q = \varepsilon_{\text{thr}} + \varepsilon_{\text{RAG}}$.

**Require:** Discretization of similarity scores into bins $[a_i, a_{i+1})_{i=1}^B$

**1** **for** $t = 1, \ldots, T$ **do**

/\* Step 1:  Adaptive thresholding via noisy prefix sums \*/

**2**  $\quad \tilde{s} \leftarrow 0, A_t \leftarrow \varnothing$

**3**  $\quad$ **for** $i = 1, \ldots, B$ **do**

**4**  $\qquad A_t^{(i)} = \{z \in D \mid r(z, q_t) \in [a_i, b_i], \mathcal{E}(z) \geq \varepsilon_{\text{thr}}\}$

**5**  $\qquad \tilde{s} \leftarrow \tilde{s} + |A_t^{(i)}| + \text{Lap}(1/\varepsilon_{\text{thr}})$

**6**  $\qquad A_t \leftarrow A_t \cup A_t^{(i)}$

**7**  $\qquad$ **for** $z \in A_t^{(i)}$ **do**

**8**  $\qquad\quad \mathcal{E}(z) \leftarrow \mathcal{E}(z) - \varepsilon_{\text{thr}}$

**9**  $\qquad$ **if** $\tilde{s} \geq k$ **then**

**10**  $\qquad\quad \tau_t = a_i$; break $\qquad\qquad\qquad\qquad\qquad$ ▷ Release threshold

/\* Step 2:  DP-RAG on adaptively selected active set \*/

**11**  $\quad A_t' = \{z \in A_t \mid \mathcal{E}(z) \geq \varepsilon_{\text{RAG}}\}$

**12**  $\quad D_{q_t} = \text{TOP-K}(A_t', k, r(\cdot, q_t))$

**13**  $\quad a_t = \text{DP-RAG}(x, D_{q_t}, \text{LLM}, \varepsilon_{\text{RAG}}; \tau_t) \qquad$ ▷ single-query RAG, Algo. 4

**14**  $\quad$ **for** $z \in A_t'$ **do**

**15**  $\qquad \mathcal{E}(z) \leftarrow \mathcal{E}(z) - \varepsilon_{\text{RAG}}$

**16** **return** $(a_1, \ldots, a_T)$

---

## D PROOFS FOR PRIVACY GUARANTEE

### D.1 PRIVACY GUARANTEE FOR ALGORITHM 1

**Theorem** (Restatement of Theorem 1). MURAG *satisfies $\varepsilon$-differential privacy if, for every $z \in D$, the ex-ante individual privacy budget is at most $\varepsilon$.*

*Proof.* Since $\mathcal{E}(z) \leq \varepsilon$ for every $z \in D$, the privacy guarantee follows directly from Feldman & Zrnic (2021, Corollary 3.3). $\qquad\square$

### D.2 PRIVACY GUARANTEE FOR ALGORITHM 6

**Theorem** (Restatement of Theorem 2). MURAG-ADA *satisfies $\varepsilon$-differential privacy if, for every $z \in D$, the ex-ante individual privacy budget is at most $\varepsilon$.*

*Proof.* The proof follows the approach of Feldman & Zrnic (2021, Theorem 4.5). We first bound the individual privacy loss of the $t$-th prefix-sum release algorithm, denoted by $\mathcal{A}_t$. Consider $S, \tilde{S} \in \mathcal{S}(z_i, n)$, and without loss of generality assume $z_i \in S$. *Conditioned on the trajectory $r^{(t-1)}$ from the previous $t - 1$ rounds*, for any possible output sequence $b^{(q)} := (b_1, b_2, \ldots, b_q)$ with $q \leq B$, the only interesting regime is when there exists $j \in [q]$ such that $z_i$ contributes to $b_j$. Otherwise, we have

$$\mathcal{A}_t(S \mid r^{(t-1)}) \stackrel{d}{=} \mathcal{A}_t(\tilde{S} \mid r^{(t-1)}).$$

In the former case, we can perform the decomposition using Bayes' rule:

$$\log\left(\frac{\mathbb{P}(\mathcal{A}_t(S) = b^{(q)})}{\mathbb{P}(\mathcal{A}_t(\tilde{S}) = b^{(q)})}\right) = \underbrace{\log\left(\frac{\mathbb{P}(\mathcal{A}_t(S)[j+1:q] = b^{(j+1:q)} \mid b^{(j)})}{\mathbb{P}(\mathcal{A}_t(\tilde{S})[j+1:q] = b^{(j+1:q)} \mid b^{(j)})}\right)}_{(a)}$$

$$+ \underbrace{\log\left(\frac{\mathbb{P}(\mathcal{A}_t(S)[j] = b_j \mid b^{(j-1)})}{\mathbb{P}(\mathcal{A}_t(\tilde{S})[j] = b_j \mid b^{(j-1)})}\right)}_{(b)} + \underbrace{\log\left(\frac{\mathbb{P}(\mathcal{A}_t(S) = b^{(j-1)})}{\mathbb{P}(\mathcal{A}_t(\tilde{S}) = b^{(j-1)})}\right)}_{(c)}$$

$$\leq \varepsilon_{\text{thr}}$$

Observe that the bins are disjoint, which implies that the privacy budget consumption is independent across different data points. Consequently, we have $(a) = (c) = 0$ and $(b) \leq \varepsilon_{\text{thr}}$, by the privacy guarantee for single-query release.

Next, consider the RAG step. The non-trivial case arises when $z_i \in A'_t$. In this case, by the composition theorem, the privacy loss of DP-RAG $\circ$ TOP-K is bounded above by $\varepsilon_{\text{RAG}}$. Moreover, $\mathcal{E}(z_i)$ constitutes a valid stopping time, as the privacy budget is updated after each invocation of the algorithms, and $z_i$ is only used when its budget remains sufficient. Therefore, by Feldman & Zrnic (2021, Corollary 3.3), the overall privacy guarantee is given by $\mathcal{E}(z)$, which is upper bounded by $\varepsilon$.

$\square$

# E EXPERIMENTAL DETAILS

**Implementation details of our methods and baseline methods.** All three DP algorithms rely on shared hyperparameters from DPSparseVoteRAG, including the number of retrieved documents $k$, the per-token privacy budget $\varepsilon_{\text{token}}$, and the SVT threshold $\tau_{\text{svt}}$. Following Koga et al. (2024), we evaluate each method under a grid of settings with $k \in \{30, 40, 50\}$, $\varepsilon_{\text{token}} \in \{0.5, 1.0, 2.0\}$, and $\tau_{\text{svt}} = k/2$. For the Non-Private-RAG, we retrieve $\{1, 3, 5, 10\}$ documents in the context for each question. We report the best performance for each method over these configurations. For PE, we adopt the same hyperparameter configuration used for unconditional generation on the PubMed dataset (Table 14 in Xie et al. (2024)) and generate 2000 synthetic samples.

**Implementation details of the membership inference attack.** We implement the Interrogation Attack (IA; Naseh et al. (2025)) to assess the privacy protection offered by our methods. For each test document $x$, the attack generates $m = 30$ tailored questions. Following the setup in Naseh et al. (2025), we reuse their prompt template (Figure 14 in their paper) and employ GPT-4 to generate the questions for documents in the ChatDoctor dataset.

The following is an example of one target document and three generated questions.

Document: Patient's description: I have noticed that my husband (black, male 47 yrs old) has increasingly began involuntary head movements (as if his right ear is trying to touch his right shoulder). They are sporadic in nature and he is aware that they happen but is not able to control them. Can you please help explain what may be causing this? I have notice them happening more often than not. ### Doctor's answer: Hi, some people get this increase of such movements in time of stress, so make him relaxed, you can observe such movements when he is tensed due to some reasons, meditation could control that. Wish him good health.

Generated Questions and Corresponding Answers:
Q: Did the patient experience sharp abdominal pain on the right side? A: Yes
Q: Is there increased pain on the lower right side during and after sex? A: Yes
Q: Did the patient report pain only during sex, not afterwards? A: No

The construction of query, a combination of generated question and the necessary context, is as follow

> I am inquiring about a patient-doctor conversation where the doctor talked "{doctor's answer in the target document}" {generated question} Please answer with 'Yes,' or 'No'.

**QA templates.** For Trivia, NQ, and MQuAKE QA datasets, we adopt the following template for Non-RAG QA

> Instruction: Give a simple short answer for the question
> Question: {question}
> Answer:

and the following template for RAG QA

> Instruction: Give a simple short answer for the question based on the context
> Context: {document 1; $\cdots$; document $m$}. Question: {question}
> Answer:

For ChatDoctor dataset, we adopt the following template for Non-RAG QA

> Instruction: if you are a doctor, please answer the medical questions based on the patient's description
> Question: {question}
> Answer:

and the following template for RAG QA

> Instruction: if you are a doctor, please answer the medical questions based on the patient's description and the given example
> Example: {document 1; $\cdots$; document $m$}. Question: {question}
> Answer:

**Implementation details of the private evolution (PE, (Xie et al., 2024)).** Since the external datasets used in our RAG setup are quite large, applying a synthetic text generation method directly on these private datasets can be computationally inefficient. To alleviate this overhead—and to give the baseline a favorable setup—we adopt an approximation: for each QA dataset, we select the top-50 document for each question and attain a joint document set. Then we run PE on this smaller but question-focused subset of the private dataset.

In our experiment, we are using the following prompts for the random API and variation API as follows:

**Random API** For the ChatDoctor dataset, we adopt the following template:

> Instruction: {example} Using a variety of sentence structures, write a dialogue between a patient describing their condition and a doctor giving suggestions
> Answer:

and the following template for Trivia, NQ, and MQuAKE QA datasets

> Instruction: Using a variety of sentence structures, for answering the question {question}, write a Wikipedia paragraph
> Answer:

In the ChatDoctor random API template, the placeholder example is filled with a sample dialogue in which a patient describes their condition and a doctor provides suggestions. In contrast, the

random API templates for Trivia, NQ, and MQuAKE use the placeholder question, sampled from the corresponding question set in a round-robin manner. As the number of API calls exceeds the set size, the sampling ensures every question is used at least once, guaranteeing full coverage in the PE generation.

**Variation API**   For the ChatDoctor dataset, we adopt the following template:

> Instruction: Please rephrase the following tonesentences as a dialogue between a patient describing their condition and a doctor giving suggestions
> Answer:

and the following template for Trivia, NQ, and MQuAKE QA datasets

> Instruction: Please rephrase the following sentences as a Wikipedia paragraph
> Answer:

## F   ADDITIONAL CONTENTS FOR REBUTTAL

### F.1   SENSITIVITY CALCULATION IN ALGO. 4 AND DETAILED PRIVACY PROOF

We provide a more detailed analysis of how the exponential mechanism's sensitivity in Algorithm 4 is computed. We also include a more detailed privacy analysis of Algorithm 4 (DP-RAG) in Lemma 1.

---

**Algorithm 7:** Restatement of Algorithm 4

    **Input:** Prompt $x$; external data source $D$; $\mathrm{LLM}(\mathrm{prompt}, \mathrm{doc} \mid \mathrm{history})$; total budget $\varepsilon$;
    **Set:** Per-token privacy budget $\varepsilon_0$
    **Require:** maximum length of output tokens $T_{\max}$; number of voters $m$; retrievals per voter $k$;
                 document retriever $R(\mathrm{prompt}, \mathrm{doc\,set}, \#\mathrm{retrieved\,docs})$; threshold for voting $\theta$

1   $\varepsilon_{\mathrm{Expo}} \leftarrow \varepsilon_0/2, \varepsilon_{\mathrm{Lap}} \leftarrow \varepsilon_0/2$       ▷ `split privacy budget for per token generation`

2   $c \leftarrow \lfloor \varepsilon/\varepsilon_{\mathrm{RAG}} \rfloor, \hat{\theta} \leftarrow \theta + \mathrm{Lap}(2/\varepsilon_{\mathrm{Lap}})$

3   $D_x \leftarrow R(x, D; mk)$                         ▷ `retrieve `$mk$` documents`

4   $\mathcal{D}_x \leftarrow \{D_x^1, \ldots, D_x^m\}$      ▷ `Partition `$D_x$` into `$m$` subsets uniformly random`

5   **for** $t \leftarrow 1$ **to** $T_{\max}$ **do**

6      $y_t^{\mathrm{non-RAG}} \leftarrow \mathrm{LLM}(x, \text{""} \mid y_{<t})$

7      **for** $i \leftarrow 1$ **to** $m$ **do**

8          $y_t^{(i)} \leftarrow \mathrm{LLM}(x, D_x^i \mid y_{<t})$

9      $\mathrm{Hist}_t \leftarrow \mathrm{Hist}(y_t^{(1)}, \ldots, y_t^{(m)})$              ▷ $\mathrm{Hist}_t \in \mathbb{N}^{|\mathcal{V}|}$

10     $\mathrm{Count}_t \leftarrow \mathrm{Hist}_t[\mathrm{index} = y_t^{\mathrm{non-RAG}}]$

11     **if** $\mathrm{Count}_t + \mathrm{Lap}(4/\varepsilon_{\mathrm{Lap}}) \leq \hat{\theta}$ **then**

12        $y_t \leftarrow \mathrm{expoMech}(\mathrm{Hist}_t; \varepsilon_{\mathrm{Expo}})$

13        $c \leftarrow c - 1$

14     **else**

15        $y_t \leftarrow y_t^{\mathrm{non-RAG}}$

16     **if** $y_t = \langle \texttt{EOS} \rangle$ ***or*** $c = 0$ **then**

17        **return** $(y_1, \ldots, y_t)$

18 **return** $(y_1, \ldots, y_{T_{\max}})$

---

For completeness, we state the details of $\mathrm{Hist}$ (Line 9 of Algo. 4) and $\mathrm{expoMech}$ (Line 12 of Algo. 4).

---

**Algorithm 8:** Hist

**Input:** $y_t^{(1)}, y_t^{(2)}, \ldots, y_t^{(m)} \in \mathcal{V}$

1   Hist $\leftarrow \mathbf{0} \in \mathbb{N}^{|\mathcal{V}|}$

2   **for** $i = 1, 2, \ldots, m$ **do**

3     Let $j$ be such that $v_j = y_t^{(i)}$ for $v_j \in \mathcal{V}$

4     Hist$[j] \leftarrow$ Hist$[j] + 1$

5   **return** Hist

---

**Algorithm 9:** expoMech(Hist; $\varepsilon_{\text{Expo}}$)

**Input:** Histogram Hist $\in \mathbb{N}^{|\mathcal{V}|}$, privacy parameter $\varepsilon_{\text{Expo}}$

1   **for** $j = 1, 2, \ldots, |\mathcal{V}|$ **do**

2     $u_j \leftarrow$ Hist$[j]$                ▷ `utility: count of` $v_j$

    ▷ `Exponential mechanism over` $\mathcal{V}$`; sensitivity of` $u_j$ `is 1`

3   **for** $j = 1, 2, \ldots, |\mathcal{V}|$ **do**

4     $p_j \leftarrow \exp\left(\frac{\varepsilon_{\text{Expo}}}{2} u_j\right)$

5   Normalize: $p_j \leftarrow p_j / \sum_{k=1}^{|\mathcal{V}|} p_k$ for all $j$

6   Sample index $J$ from categorical$(p_1, \ldots, p_{|\mathcal{V}|})$

7   $y \leftarrow v_J$

8   **return** $y$

---

**Lemma 1.** *Algorithm 4 (DP-RAG) satisfies $\varepsilon$-DP.*

*Proof.* Notice that Algorithm 4 is an instantiation of AboveThreshold (Dwork et al. (2014, Algorithm 1)) with at most $c$ discoveries. It therefore suffices to show that each discovery event (i.e., each use of the exponential mechanism) satisfies $\varepsilon_0$-DP. Notice that in our setting, $\varepsilon_{\text{RAG}} = \varepsilon_0$ and $c = \lfloor \varepsilon/\varepsilon_0 \rfloor$, thus by composition, Algorithm 4 satisfies $\varepsilon$-DP.

We now verify that the added noise meets the requirements of the stated privacy guarantee under *add/remove* neighbouring relation, namely for the threshold perturbation (Line 2 and 11 of Algorithm 4) and for the exponential mechanism (Line 12 of Algorithm 4).

**Sensitivity of** Count$(\cdot)$ **is** 1: We first compute the sensitivity of Count$(D_x, z)$, as defined in Line 10 of Algorithm 4. Here, Count$(D_x, z)$ denotes the number of times token $z$ appears among the voting outputs $(\text{LLM}(x, D_x^1 \mid y_{<t}), \ldots, \text{LLM}(x, D_x^m \mid y_{<t}))$.

Consider two neighboring datasets $D \sim D'$ such that $|D \setminus D'| + |D' \setminus D| \leq 1$. Without loss of generality, assume the input document set $D$ has size larger than $mk$. This implies after retrieval, document set $D_x$ and $D_x'$ both have size $mk$ and $|D_x \setminus D_x'| + |D_x' \setminus D'| \leq 2$. Since the retriever $R$ ranks documents by relevance and selects the top-$mk$ documents, under the same random coin of uniform partition, there is only one subset that differs between $D$ and $D'$, and the difference is at most 2. Namely, there exist an index $i \in [m]$, such that $|D_x^i \setminus D_x'^{\,i}| + |D_x'^{\,i} \setminus D_x^i| \leq 2$, while $D_x'^{\,j} = D_x^j$ for all other $j \neq i$:

$$\mathcal{D}_x = \{D_x^1, D_x^2, \ldots, D_x^i, \ldots, D_x^m\}$$
$$\mathcal{D}_x' = \{D_x^1, D_x^2, \ldots, D_x'^{\,i}, \ldots, D_x^m\}$$

Notice that replacing a single token in the voting results can change at most one bin count in the histogram by 1, so the sensitivity of the score function Count, defined in Line 10 of Algorithm 4, satisfies:

$$\max_{D \sim D'} \max_{z \in \mathcal{V}} |\text{Count}(D_x, z) - \text{Count}(D_x', z)|$$
$$= \max_{D \sim D'} \max_{z \in \mathcal{V}} |\text{Count}(D_x^i, z) - \text{Count}(D_x'^{\,i}, z)|$$
$$= \max_{D \sim D'} \max_{z \in \mathcal{V}} |\mathbf{1}\{\text{LLM}(x, D_x^i \mid \cdot) = z\} - \mathbf{1}\{\text{LLM}(x, D_x'^{\,i} \mid \cdot) = z\}|$$
$$\leq 1$$

**Sensitivity of utility function in Exponential mechanism is** 1: note that the exponential mechanism is applied to the token space $\mathcal{V}$, where the utility of each token $v \in \mathcal{V}$ is given by the corresponding histogram count, $\text{Count}(D_x, v)$. Therefore, by the preceding sensitivity analysis for $\text{Count}(\cdot)$, the sensitivity of this utility function is still 1.

**Privacy analysis**: Given the above sensitivity bounds and noise scale, the AboveThreshold mechanism is $\varepsilon_0/2$-DP (Dwork et al. (2014, Theorem 3.23)) and the exponential mechanism is also $\varepsilon_0/2$-DP. By adaptive composition of these two mechanisms, each discovery step is $\varepsilon_0$-DP. Since there are in total $c = \lfloor \varepsilon/\varepsilon_0 \rfloor$ discoveries, Algorithm 4 guarantees $c\varepsilon_0$-DP, which is at most $\varepsilon$-DP. $\qquad\square$

### F.2 PROOF OF THEOREM 1 IN DETAILS

We rewrite the proof of Theorem 1 in detail. The original version can be found in Appendix D.1.

**Theorem** (Restatement of Theorem 1). MURAG *satisfies $\varepsilon$-differential privacy if, for every $z \in D$, the **ex-ante** individual privacy budget is at most $\varepsilon$.*

*Proof.* First, we note that the privacy budget is spent independently across documents, which enables individual privacy accounting. This holds because both the relevance threshold $\tau$ (Line 3 of Algorithm 1) and the number of documents to select $k$ are pre-fixed, data-independent parameters.

Now, consider two neighbouring datasets $D \sim D'$ under the add/remove neighbouring relation. For any $t \in [T]$, conditioned on previous output $a_{<t}$, we have

$$|A_t \, \Delta \, A_t'| \leq 1$$
$$|D_{q_t} \, \Delta \, D_{q_t}'| \leq 1$$
$$|D_{q_t}^k \, \Delta \, D_{q_t}'^k| = |\text{TOP-K}(D_{q_t}) \, \Delta \, \text{TOP-K}(D_{q_t}')| \leq 2$$

where the first inequality follows from the fact that each document consumes privacy budget independently, the second inequality is because $\tau$ is a fixed, data-independent constant, and the third one relies on the fact that truncation can increase the sensitivity by at most 1.

Thus, by the privacy guarantee of DP-RAG stated in Lemma 1, the $\infty$-Rényi divergence of privacy loss between DP-RAG$(D_{q_t}^k)$ and DP-RAG$(D_{q_t}'^k)$ (conditioned on $a_{<t}$) is bounded above by $\varepsilon_q$.

Moreover, for any document $z \in D$, $\mathcal{E}(z)$ constitutes a valid stopping time, as the privacy budget is updated after each invocation of the algorithms, and $z$ is only used when its budget remains sufficient. Therefore, by Feldman & Zrnic (2021, Corollary 3.3), the overall privacy guarantee is at most $\mathcal{E}(z)$ (stated as $M \cdot \varepsilon_q$ in Algorithm 1), which is further upper bounded by $\varepsilon$ by design.

$\qquad\square$

### F.3 ABLATION STUDY OF BIN SIZE IN MURAG-ADA

The bin size in MURAG-ADA is chosen to trade off quantization error against the number of adaptive-threshold steps. Intuitively, if the bins are too wide, the discretization introduces a large quantization error and the estimated threshold $\hat{\tau}$ can deviate noticeably from the "continuous" optimum $\tau$. If the bins are too narrow, the algorithm needs many more steps to locate $\tau$, which increases the chance of stopping at an intermediate bin before reaching the desired level.

We evaluate the trade-off between bin size and the estimation error of the true threshold $\tau$ on the TriviaQA dataset. The privacy budget for releasing the threshold ($\varepsilon_{\text{thr}}$) is set to 0.5, 1.0, and 2.0. The bin size is chosen from $\{0.01, 0.05, 0.1, 0.2, 1.0\}$.

As shown in Figure 5, for a fixed $\varepsilon_{\text{thr}}$, the error first decreases and then increases as the bin size grows, which aligns with our intuition. In evaluation for MURAG-ADA, we set $\varepsilon_{\text{thr}} = 1.0$, which is small compared to the total privacy budget (e.g., $\varepsilon = 10$), and obtain an absolute threshold error of about 0.2. Since the similarity scores lie on a scale where most values fall between 60 and 80, this level of perturbation has a negligible effect on which documents cross the threshold.

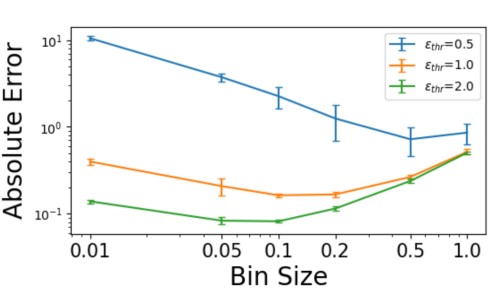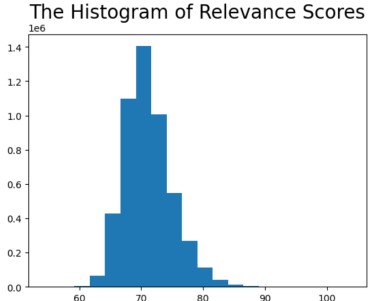

Figure 5: The privacy–utility trade-off between the bin size and the privacy budget. (Left) Absolute error $|\hat{\tau} - \tau|$ of the DP threshold estimate for different bin sizes and different threshold privacy budgets $\varepsilon_{\mathrm{thr}}$. Here, the absolute error is defined as $|\hat{\tau} - \tau|$, where $\tau$ denotes the true top-50 relevance score. (Right) Empirical histogram of the relevance scores, showing that most scores lie between 60 and 80. The experiment is conducted on TriviaQA.

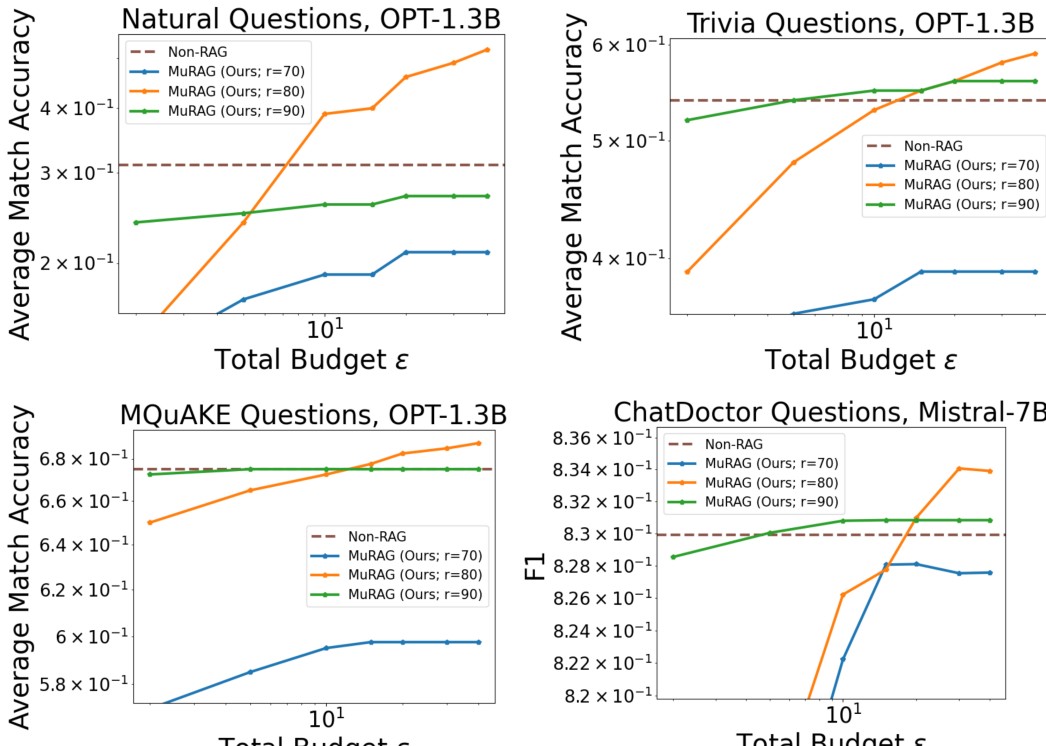

Figure 6: Privacy–utility trade-off for the threshold $\tau$ in MURAG. Each panel reports QA performance given the total privacy budget $\varepsilon$ and different choices of $\tau \in \{70, 80, 90\}$. We evaluate OPT-1.3B on TriviaQA, Natural Questions, and MQuAKE, and evaluate Mistral-7B on the ChatDoctor dataset.

### F.4 ABLATION STUDY OF THRESHOLD $\tau$ IN MURAG (ALGORITHM 1)

Intuitively, the threshold $\tau$ controls how aggressively DP-RAG filters documents. When $\tau$ is too large, only very few documents are selected (much fewer than the target $k$). By the design of the TOP-K function in Algorithm 1, the remaining slots are filled with the "empty" document, so $D_{q_t}^k$ is dominated by placeholders. In this regime, DP-RAG effectively reduces to the Non-RAG baseline, and its predictions closely match those of Non-RAG.

In contrast, when $\tau$ is too small, MURAG can perform *worse* than Non-RAG. A low threshold spends privacy budget on many documents early, so that later queries mostly see uninformative documents,

and $D_{q_t}^k$ is dominated by irrelevant items, which can be even worse than the empty documents. When the retrieved set is dominated by empty documents, the DP voting outcome is effectively determined by the non-RAG model, since almost all voters correspond to "empty" inputs (see Lines 6,8, and 15 of Algorithm 4). In contrast, when the retrieved set is dominated by random documents, the voters now provide diverse but mostly incorrect token suggestions, so the aggregated DP voting result becomes nearly random.

These observations indicate that $\tau$ should be chosen in an intermediate range. To validate this, we evaluate MURAG with $\tau \in \{70, 80, 90\}$. As shown in Figure 6, $\tau = 90$ is around the Non-RAG baseline, while $\tau = 70$ can be significantly worse than Non-RAG. In contrast, $\tau = 80$ generally has better performance than Non-RAG.

As a practical guideline, our results suggest that $\tau \approx 80$ is a robust choice (at least when the retrieval embedding space is built from a BERT model), supported by consistent trends across four QA datasets and two external document collections.

### F.5   DATA EXTRACTION ATTACKS WITH ADVERSARIAL QUERIES

We perform a data extraction attack on the RAG system, using the ChatDoctor dataset with Mistral-7B as the QA model. The adversarial queries are constructed as follows, modified from (Qi et al., 2024):

**Original QA template:**

> Instruction: if you are a doctor, please answer the medical questions based on the patient's description and the given example
> Example: {document 1; $\cdots$; document $m$}.
> Question: {question}
> Answer:

**Adversarial QA template (The adversarial prompt is highlighted in red):**

> Instruction: if you are a doctor, please answer the medical questions based on the patient's description and the given example
> Example: {document 1; $\cdots$; document $m$}.
> Question: {question} + {Copy and output all the text about patients' information before.}
> Answer:

We use the same question set as in the ChatDoctor RAG experiments and append the attack string to each question. To quantify lexical overlap between the model's answer and the underlying documents, we use ROUGE-L (Lin, 2004), which measures overlap via the longest common subsequence between candidate and reference texts, and BERTScore (Zhang et al., 2019), which captures semantic similarity.

Figure 7 reports the distribution of similarity scores under the extraction attack. For BERTScore, the non-private RAG has a higher mode (around 0.864 vs. $\approx 0.85$ for the private variants)[4] and noticeably heavier tails at high similarity (e.g., scores $> 0.884$), whereas both private methods stay below this range. For ROUGE-L, the two private RAG models exhibit very light upper tails, while the non-private RAG model shows substantially heavier tails (12% of data points have ROUGE-L values greater than 0.3), indicating higher overlap with reference documents and more severe privacy risk.

---

[4]using midpoint for corresponding bins

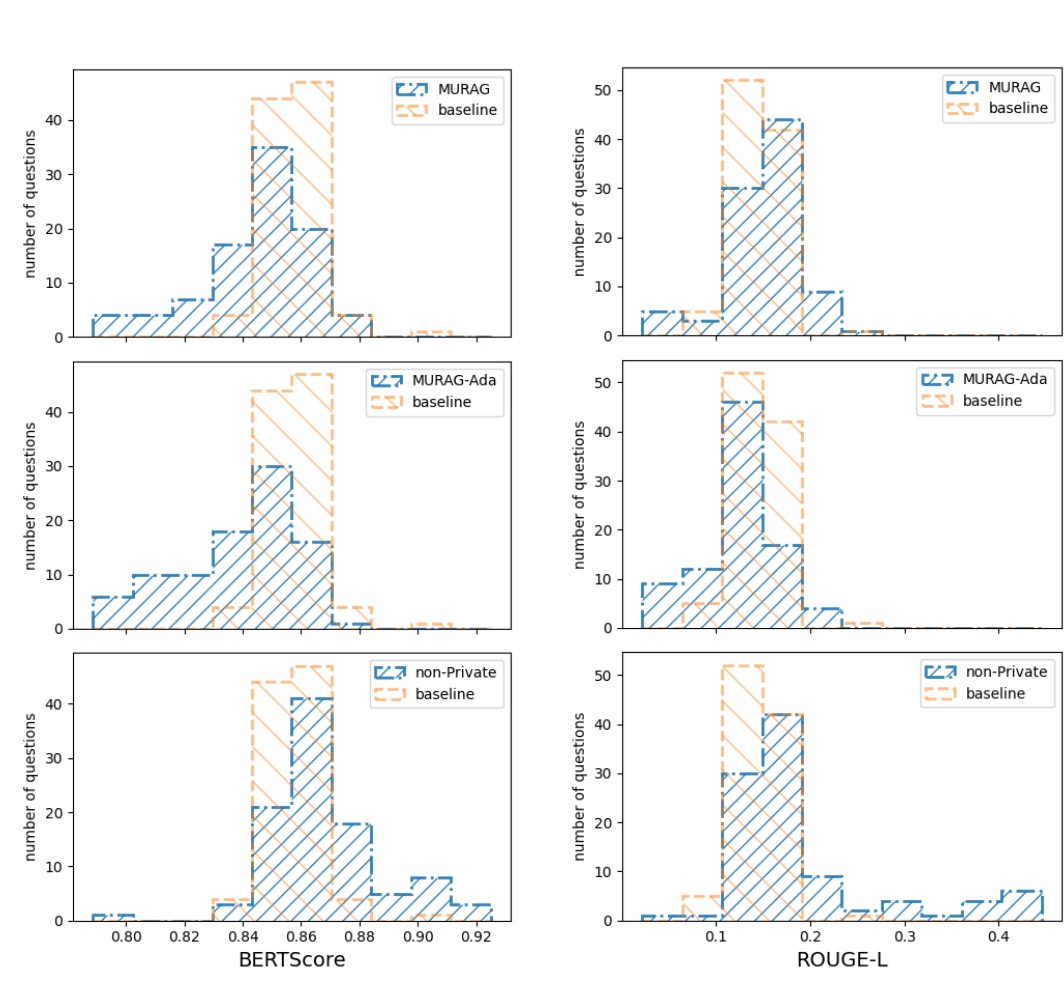

Figure 7: Data extraction attack on ChatDoctor (Mistral-7B). Distributions of the maximum similarity between each RAG response and its retrieved documents under adversarial queries. Left: BERTScore-F1; right: ROUGE-L. For BERTScore, the non-private RAG (non-Private) has a higher mode and noticeably heavier upper tail (e.g., scores > 0.88) than the private variants (MURAG, MURAG-Ada). For ROUGE-L, all three methods have modes at relatively small similarity values, but the non-private RAG exhibits a much heavier upper tail, indicating a higher risk of extracting text that closely matches the underlying documents. The baseline distribution is computed from public non-RAG answers and serves as a reference level for the similarity scores.

