# OpenReview forum: "Private-RAG: Answering Multiple Queries with LLMs while Keeping Your Data Private"
_ICLR.cc/2026/Conference — Submitted to ICLR 2026_

### Official Review · Reviewer_mpC4 · 2025-10-28

**Soundness:** 2
**Presentation:** 2
**Contribution:** 3
**Rating:** 4
**Confidence:** 5

**Summary:**

The paper draws the attention that in a real-world, single-query DP RAG cannot meet the privacy requirements in multiple round query settings as the privacy budget will accumulate. A large privacy budget is needed to reach acceptable utility. To encounter the challenge, the authors proposed a revised framework using a privacy filter for each document. Experiments were conducted to support the main argument.

**Strengths:**

1. The paper identified current challenge on privacy RAG where sufficiently large $\epsilon$ is needed to achieve reasonable utility, which is an important conclusion. The illustration of motivation is important to following research.
2. The formulation of the research question is clear.
3. Abundant experiments are conducted to study the effectiveness of multiple-round query.

**Weaknesses:**

1. In Konga et al., the guarantee is stated in $(\epsilon,\delta)$-DP, but this paper reports RDP. Please clarify the conversion pathway and assumptions. In particular, walk readers through how the moments accountant (or equivalent) maps the mechanism’s $(\epsilon,\delta)$-DP bounds to $(\alpha,\epsilon_\alpha)$-RDP, and specify where Appendix C/D supply the exact orders $\alpha$, composition steps, and the final back-conversion (if any). Please see more in my question.
2. The tuning of $\gamma$ appears critical, especially because a document is excluded once its budget is exhausted. Please analyze utility--privacy trade-offs across $\gamma$, provide guidance for selection (e.g., via validation curves or heuristics), and discuss robustness (variance across runs/datasets).
3. The current evaluation relies on privacy budgets and membership-inference attacks (MIA), which are indirect. For stronger external validity, include empirical extraction tests: issue adversarial queries and measure whether private content (documents/strings) is retrieved or generated. Reporting concrete leakage rates alongside MIA would materially strengthen the privacy claims.

Minor:
- Typo: `R'enyi` --> `Rényi`.
- Since Algorithm 5 is referenced in the main text, consider moving it from the appendix to the main body to improve readability.

**Questions:**

1. What does notation $S\Delta\tilde{S} $mean?

2. How is the utility guaranteed if a document is excluded. Say one document contains a lot of privacy information and it is frequently visited, which is common in the real-world, how would the following queries be correct if it is excluded?

 think these should be included in the main paper. My confusion was not resolved by reading the main paper so I turned to check the Appendix. You start with a single query DP and replace the limited domain with an exponential mechanism in the private token generation step. How do you calculate the sensitivity? Exponential mechanism is $\epsilon$-DP and so as the conclusion in D.1. Then how is it converted to RDP in D.2. I think (b)<$\epsilon$ in D.2 is based on D.1. But how is the conclusion in D.1 derived? Corollary 3.3 is based on $\epsilon,\alpha$-DP, how do you make it to the $\epsilon$-DP? What value of $\alpha$ do you select for RDP.

3. Why other baselines are not presented in Figure 3 (Right).

**Details Of Ethics Concerns:**

Datasets and models are publicly available and allowed for non commercial case. No privacy data are included.

---

> ### Author Response · Authors · 2025-11-27
>
> Thank you for acknowledging the importance of our conclusion and the comprehensiveness of our empirical evaluation. We address your questions below:
>
>
> > Weakness 1: "In Konga et al., the guarantee is stated in $(\epsilon,\delta)$-DP ...Please see more in my question." + Question: "Corollary 3.3 is based on $\epsilon,\alpha$-DP, ... What value of $\alpha$ do you select for RDP."
>
> Thanks for your question. The $(\varepsilon, \delta)$-DP guarantee in Konga et al. comes from the LimitedDomain mechanism in their Algorithm 2, which yields $(\varepsilon, \delta)$-DP. In our work, we replace this mechanism with the exponential mechanism, so all of our algorithms satisfy pure $\varepsilon$-DP. For the conversion between $(\alpha, \varepsilon)$-RDP and $\varepsilon$-DP, we set $\alpha = \infty$, as $\varepsilon$-DP is equivalent to $(\infty, \varepsilon)$-RDP.
>
>
> > Weakness 2: "The tuning of $\gamma$...and discuss robustness (variance across runs/datasets)."
>
> Thanks for your question. As a clarification, we believe that the $\gamma$ you mentioned corresponds to the threshold $\tau$ used in Algorithm 1 (MURAG).
>
>
> We first state the heuristics for picking the appropriate $\tau$ parameter (we use the same notations as in Algorithm 1):
>
> - **$\tau$ can not be too large**. If so, the generated answer from DP-RAG will be mostly the same as that of Non-RAG (i.e., RAG with the empty documents). This is because thresholding at a very high relevance score can only filter in a few documents, leading to the pre-screened document set $D_{q_t}$ containing far fewer documents than $k$. This further causes the refined retrieved document set $D_{q_t}^k$ to be dominated by “empty” documents, as the $top\text{-}K$ function adds them until the total number of retrieved documents reaches $k$. As a result, the generated answer from DP-RAG would be largely the same as that of Non-RAG, because the voters inside DP-RAG mostly receive empty input documents (see Lines 5–15 of Algorithm 4 for more details).
>
> - **$\tau$ can not be too small**. In this case, MURAG's performance can even be worse than the Non-RAG. This is because a low relevance score threshold allows more irrelevant documents into the pre-screened document set $D_{q_t}$, causing $D_{q_t}^k$ to be dominated by irrelevant documents. As a result, the voting outcome from DP-RAG becomes incorrect, which is even worse than voting without context (i.e., Non-RAG). In addition, privacy budgets for currently irrelevant documents are also consumed, potentially making them unavailable for future actual relevant queries, thereby degrading the overall performance of the RAG system.
>
> This heuristic suggests that an appropriate value of $\tau$ should be some value in the middle. Our further ablation study of different choices of $\tau$ validates this intuition. In our experiments, we evaluated MURAG using three values of $\tau \in$ {70, 80, 90}. Figures 2 and 3 report MURAG's best performance across these three values for each privacy budget $\varepsilon$ shown on the x-axis. Additionally, we further report the performance of MURAG with each $\tau$ in Figure 6 of Appendix F.4. As observed, a large $\tau$ (e.g., 90) results in performance almost around that of Non-RAG, while a small $\tau$ (e.g., 70) performs significantly worse. In contrast, MURAG ($\tau = 80$) generally outperforms Non-RAG.
>
> As a practical recommendation, we hypothesize that $\tau = 80$ is generally a good choice, at least when the embedding space (for retrieval) in RAG is constructed using the BERT model. This hypothesis is supported by the consistently better performance of $\tau = 80$ across four question datasets and two external document sets.
>
>
> > Weakness 3: "The current evaluation relies on .... Reporting concrete leakage rates alongside MIA would materially strengthen the privacy claims."
>
> To address this question, we added a data-extraction attack experiment using adversarial queries (Appendix F.5). Specifically, we evaluated Mistral-7B on the ChatDoctor dataset and measured information leakage using BERTScore-F1 and ROUGE-L-F1 between the generated answers and the retrieved documents. As shown in Figure 7 (Appendix F.5), for BERTScore, the non-private RAG demonstrates a higher mode and a heavier upper tail (e.g., BERTScores $> 0.88$) than the private RAG variants (MURAG and MURAG-Ada). For ROUGE-L, all three methods exhibit modes at relatively low similarity values; however, the non-private RAG again shows a significantly heavier upper tail ($12\%$ of data points have ROUGE-L score larger than $0.3$), indicating a greater likelihood of close matches with the underlying documents. **These results suggest that MURAG and MURAG-Ada substantially reduce extraction risk compared to the non-private RAG.**

---

> ### Author Response · Authors · 2025-11-27
>
> > Minor Weakness
>
> We have corrected the typos you pointed out. We agree that including Algorithm 5 in the main text would improve readability and have moved it there in the revised version (please check the change log in the general response).
>
>
>
>
> > Question 1: "What does this notation $S \Delta \tilde S$ mean?"
>
> $S \Delta \tilde{S}$ means the symmetric difference between sets $S$ and $\tilde{S}$. Namely, $S \Delta \tilde{S} = (S \/ \tilde{S}) \cup ( \tilde{S} \/ S)$. We have included this definition in the notation section of the revised version (Lines 78-79).
>
>
> > Question 2: "How is the utility guaranteed if a document is excluded... how would the following queries be correct if it is excluded?
>
> We appreciate your insightful question. In this case, accurate answers to all these questions would violate the DP definition for a meaningful privacy budget (in a way that is reminiscent of the celebrated Dinur-Nissim lower bound [1]). This applies to any DP mechanisms, and ours is no exception.  Information-theoretically, no good privacy-utility tradeoff exists for these queries. On the other hand, in standard DP-RAG without individual accounting, a single heavily-used document can drain the *global* privacy budget and shut down the whole system. In our approach, the impact is localized: only queries that rely on that specific document are affected, while we can still answer many other RAG queries that do not use it.
>
> We also note that when a document is repeatedly accessed by semantically identical or very similar queries, we can reuse previously released answers via caching. This is purely a post-processing step and therefore does not incur any additional privacy cost. (See the discussion section “Leveraging Historical QA” for more details.)
>
> [1] *Dinur, Irit, and Kobbi Nissim. "Revealing information while preserving privacy." Proceedings of the twenty-second ACM SIGMOD-SIGACT-SIGART symposium on Principles of database systems. 2003.*
>
> > Question: "You start with a single query DP and replace the limited domain with an exponential mechanism...I think (b)< $\varepsilon$ in D.2 is based on D.1. But how is the conclusion in D.1 derived? "
>
> Thank you for your question. We have added detailed sensitivity calculations in Appendix F.1, along with a detailed proof of Theorem 1—namely, the conclusion stated in Appendix D.1—in Appendix F.2 for your reference.
>
> > Question 3: "Why other baselines are not presented in Figure 3 (Right)."
>
> The goal of Figure 3 (right) is to illustrate that even at $\varepsilon = 10$, the privacy guarantee of MURAG and MURAG-ada remains strong. As shown in Figures 2 and 3 (left), Naive multi-RAG and Private Evolution already suffer from poor utility at $\varepsilon = 10$. Thus, even if these methods were to offer stronger privacy protection, their overall privacy-utility trade-off would still be unfavorable.

---

> > ### Comment · Reviewer_mpC4 · 2025-11-27
> >
> > Thank you very much for the detailed rebuttal. I have also read the other reviewers’ comments and your responses. I appreciate the clarifications regarding my questions and the weaknesses I raised.
> >
> > Apologies for my earlier mistake — in Weakness 2, I indeed meant to refer to $\tau$, not $\gamma$.
> >
> > I have updated my rating accordingly.
> >
> > Regarding Question 3:
> > I believe the writing in the evaluation section could be improved in future revisions. For example, Figures 2, 3, and 7 could potentially be reorganized or combined in a way that highlights two key points more clearly:
> >
> > 1. MURAG(-ada) demonstrates a strong privacy–utility trade-off, both in terms of privacy budgets and extraction-attack robustness.
> >
> > 2. Prior DP-based methods do not achieve a comparable trade-off; at the same privacy budget, MURAG provides significantly better utility.
> >
> > I hope these suggestions are helpful for improving the clarity of the evaluation narrative.

---

> ### Author Response · Authors · 2025-12-03
>
> We are glad that our clarifications have addressed your questions and concerns, and we greatly appreciate your decision to raise the score to 6.
>
> We also thank you for your suggestions to improve the clarity of the evaluation narrative and have incorporated them into the corresponding paragraphs in “Results on privacy-sensitive applications” (lines 412–427) and the related content (lines 455–457). We hope these revisions more clearly highlight the significantly improved privacy–utility trade-off achieved by our methods.

---

### Official Review · Reviewer_hg6W · 2025-10-28

**Soundness:** 2
**Presentation:** 2
**Contribution:** 2
**Rating:** 2
**Confidence:** 4

**Summary:**

This paper proposes two multi-query differentially private RAG algorithms (MURAG and MURAG-ADA) that improve privacy budget efficiency through per-document budget management and threshold-based screening mechanisms. The main contribution lies in engineering design optimization, extending single-query DP-RAG to multi-query scenarios, though technical innovation is limited.

**Strengths:**

1. The paper addresses the relevant problem of privacy protection in multi-query RAG systems.
2. The experimental evaluation covers multiple LLM models and includes privacy attack assessments.

**Weaknesses:**

1. The evaluation focuses primarily on privacy budget consumption while providing insufficient analysis of system complexity costs introduced by per-document budget management. Maintaining individual states for large numbers of documents may introduce significant computational overhead and storage requirements.
2. The technical contribution is mainly at the engineering design level with relatively limited core algorithmic innovation. While per-document budget management and threshold screening are effective, they primarily represent reasonable combinations of existing techniques.
3. The experimental scale is relatively limited, with the largest model being 7B parameters and a modest number of test questions, which may not adequately validate the method's performance in large-scale real-world deployments.
4. The threshold setting strategies have some limitations. Fixed threshold selection lacks clear guiding principles, while adaptive thresholds require additional privacy budget consumption, and their net benefits need further verification.
5. The practical application demand for multi-query differential privacy RAG may not be as widespread as described in the paper, as most RAG systems likely employ alternative privacy protection strategies or handle non-sensitive data.

**Questions:**

1. What would be the total system costs of per-document state management in real systems with millions of documents?
2. Could you provide a more comprehensive cost-benefit analysis that includes factors such as system complexity?
3. How can service quality be maintained when popular documents exhaust their privacy budgets?
4. Are there more principled methods for threshold parameter selection?

---

> ### Author Response · Authors · 2025-11-27
>
> We appreciate the reviewer's comments and feedback. We address your questions below:
>
> > **(Weakness 1 and Question 1, 2)** regarding "system complexity."
>
> Thank you for raising this point. *The additional state introduced by per-document accounting is very small*: we only store a single scalar privacy budget $\mathcal E (z)$ per document. This adds one additional float per document, *which is negligible compared to the space needed to store the raw documents and their embeddings (which is around 768 floats per document)*. Computationally, updating the remaining privacy budget is lightweight: for each retrieved document, we perform an O(1) update (a lookup, one subtraction, and one comparison) in a per-document state table (e.g., a hash map keyed by document ID). These updates are independent across documents and can be parallelized over the retrieved set.
>
>
> > **(Weakness 2)** Clarification of novelty
>
> We appreciate the reviewer’s recognition of the effectiveness of our algorithm. However, we respectfully disagree with the assertion that our contribution is purely at the engineering level. **For clarity, we have provided a more detailed discussion of our technical contributions in the general response, as well as additional technical details in the revised Appendices F.1 and F.2.** We hope these clarifications help to make our novel technical contributions more evident.
>
>
> > **(Weankess 3)** "The experimental scale is relatively limited....performance in large-scale real-world deployments."
>
> We believe our model size is representative of current practice in the academic literature. Comparable model scales are routinely used in recent work on synthetic data generation, private scaling laws, retrieval-augmented generation (RAG), and language model architectures, as demonstrated in publications at top-tier machine learning venues [1–6]. We highlight their base models in boldface for clarity.
>
>
>
> [1] Vinod, Vishnu, Krishna Pillutla, and Abhradeep Guha Thakurta. “InvisibleInk: High-Utility and Low-Cost Text Generation with Differential Privacy.” *NeurIPS 2025*. (**TinyLLaMA 1.1B, LLaMA3.2-1B, LLaMA3 8B**)
>
> [2] Ramesh, Krithika, et al. “Evaluating Differentially Private Synthetic Data Generation in High-Stakes Domains.” *Findings of ACL: EMNLP 2024*. (**LLaMA-1.3B, BioMistral-7B**)
>
> [3] McKenna, Ryan, et al. “Scaling Laws for Differentially Private Language Models.” *ICML 2025*. (**Bert series 4.5M ~ 778M**)
>
> [4] Li, Xinze, et al. “RAG-DDR: Optimizing Retrieval-Augmented Generation Using Differentiable Data Rewards.” *ICLR 2025*. (**MiniCPM-2.4B** and **Llama3-8B**)
>
> [5] Gao, Jingsheng, et al. “SmartRAG: Jointly Learn RAG-Related Tasks From the Environment Feedback.” *ICLR 2025*. (**Flan-T5-Large 0.8B and LLaMA-2 7B**)
>
> [6] Muennighoff, Niklas, et al. “OLMoE: Open Mixture-of-Experts Language Models.” *ICLR 2025*. (**OLMoE-1B–7B**)
>
>
>
>
> > **(Question 3)** "How can service quality be maintained when popular documents exhaust their privacy budgets?"
>
> This depends on the specific problem setting. For example, when newly issued queries are semantically similar to past queries, we can reuse previously released answers via caching (please refer to the discussion section Leveraging Historical QA). This is purely a post-processing step and therefore does not incur any additional privacy cost.

---

> ### Author Response · Authors · 2025-11-27
>
> > **(Weakness 4 and Question 4)**
>
> **(1) Regarding guiding principle for $\tau$ in MURAG:**
>
> We first provide the intuition behind choosing $\tau$. The threshold $\tau$ controls how aggressively DP RAG filters documents. If $\tau$ is too large, very few documents are selected, and the remaining slots are filled with "empty" placeholders, so DP-RAG essentially collapses to the Non-RAG baseline that conditions on empty documents. Conversely, suppose $\tau$ is too small. In that case, MURAG spends privacy budget on many early documents, leaving later queries with mostly uninformative or irrelevant items in $D_{q_t}^k$ (Line 6 in Algorithm 1), which can perform even worse than conditioning on empty documents.
>
> In our current work, the fixed threshold $\tau$ is treated as a standard hyperparameter and is chosen via grid search. We provide an additional ablation study of this hyperparameter in the revised Appendix F.4. Figure 6 reports the performance of MURAG with $\tau \in$ {70, 80, 90}. As observed, MURAG with $\tau = 90$ performs similarly to Non-RAG, while MURAG with $\tau = 70$ can be much worse than Non-RAG. In contrast, MURAG with $\tau = 80$ consistently outperforms Non-RAG. This also aligns with the experimental results in Figures 2 and 3, suggesting that $\tau = 80$ is a generally good choice for MURAG.
>
> As a practical recommendation, we hypothesize that $\tau = 80$ is generally a good choice, at least when the embedding space (for retrieval) in RAG is constructed using the BERT model. This hypothesis is supported by the consistently better performance of $\tau = 80$ across four question datasets and two external document sets.
>
>
> **(2) Adaptive $\tau$ only needs a modest privacy budget**
>
> For the adaptive-threshold variants, $ \text{MURAG-Ada} $, we allocate an additional privacy budget, $\varepsilon_{\mathrm{thr}}$, to estimate and release a data-dependent threshold. However, **the allocated budget $\varepsilon_{\mathrm{thr}} = 1$ is small relative to the total budget (e.g., $\varepsilon = 10$)**. As shown in the ablation study (Figure 5(left), Appendix F.3), this modest privacy budget already yields an absolute error of 0.2, which has a negligible effect on retrieval precision, as the similarity scores mostly lie between 60 and 80 (Figure 5 (right)).
>
>
> **(3) Benefits of our thresholding strategies**
>
> For a technical perspective on the benefits of thresholding, please refer to our general response, Points 2 and 3. Empirically, MURAG and MURAG-Ada already achieve significant performance improvements over the baseline Naive-Multi-RAG methods, as shown in Figure 2. Moreover, the adaptive threshold further yields substantial performance gains over MURAG on correlated questions (MQuAKE), as discussed in Section 4.3 and illustrated in Figure 2 (Row 3).

---

> ### Author Response · Authors · 2025-11-27
>
> > **(Weakness 5)** "The practical application demand for multi-query differential privacy RAG ...RAG systems likely employ alternative privacy protection strategies or handle non-sensitive data."
>
> * Regarding ``non-sensitive data''
>
> We respectfully disagree with the argument that retrieval-augmented generation (RAG) is primarily used only on non-sensitive corpora. While many deployed RAG systems do operate on public or low-risk data, there is a growing number of practical applications in which the retrieved documents are clearly sensitive (e.g., enterprise databases, medical or legal records, and customer support histories). Our work is motivated precisely by such settings. A clear example is the ChatDoctor dataset we use, which has become a widely adopted benchmark for evaluating privacy leakage in RAG [7–10], as it contains patient–doctor conversations involving sensitive information. Concurrently, several recent studies have examined data leakage and privacy risks in RAG systems [11–14], indicating that privacy-preserving RAG is an active and important area of research.
>
> * Regarding “alternative privacy protection strategies”:
>
> We agree that many alternative privacy protection strategies exist (e.g., access control, heuristic filters, empirical attack evaluations). *However, most of these approaches provide only empirical or attack-specific protection. They lack provable worst-case guarantees and have been shown to be vulnerable to adaptive or arbitrary adversaries (see, e.g., [15]).* In contrast, **differential privacy (DP) offers a provable, information-theoretic privacy guarantee that holds against a wide range of attack strategies [16].** This is particularly important in high-stakes domains such as healthcare [17] and compliance-sensitive enterprise applications [18], where ad hoc defenses may be insufficient. Prior work has, in fact, highlighted DP as a promising tool for supporting compliance with regulations such as the General Data Protection Regulation (GDPR) [19] and guidance from the National Institute of Standards and Technology (NIST) [20–22]. Our paper should be viewed in this context: it provides a DP-based, provable privacy guarantee for multi-query RAG systems in settings where such strong guarantees are necessary.
>
> [7] Li, Yunxiang, et al. "Chatdoctor: A medical chat model fine-tuned on a large language model meta-ai (llama) using medical domain knowledge." *Cureus* (2023).
>
> [8] Chen, Yufei, et al. "Fine-Grained Privacy Extraction from Retrieval-Augmented Generation Systems via Knowledge Asymmetry Exploitation." *arXiv preprint arXiv:2507.23229* (2025).
>
> [9] Wang, Haoran, et al. "Privacy-aware decoding: Mitigating privacy leakage of large language models in retrieval-augmented generation." *arXiv preprint arXiv:2508.03098* (2025).
>
> [10] Jiang, Changyue, et al. "Rag-thief: Scalable extraction of private data from retrieval-augmented generation applications with agent-based attacks." *arXiv preprint arXiv:2411.14110* (2024).
>
> [11] Zeng, Shenglai, et al. "The good and the bad: Exploring privacy issues in retrieval-augmented generation (rag)." *ACL* (2024). 2024.
>
> [12] Qi, Zhenting, et al. "Follow My Instruction and Spill the Beans: Scalable Data Extraction from Retrieval-Augmented Generation Systems." *ICLR* (2025)
>
> [13] Rathod, Vishal, et al. "Privacy and security challenges in large language models." *2025 IEEE 15th Annual Computing and Communication Workshop and Conference (CCWC)*. IEEE, 2025.
>
> [14] Naseh, Ali, et al. "Riddle me this! stealthy membership inference for retrieval-augmented generation." *arXiv preprint arXiv:2502.00306* (2025).
>
> [15] Nasr, Milad, et al. "The attacker moves second: Stronger adaptive attacks bypass defenses against llm jailbreaks and prompt injections." *arXiv preprint arXiv:2510.09023* (2025).
>
> [16] Wood, Alexandra, et al. "Differential privacy: A primer for a non-technical audience." *Vand. J. Ent. & Tech. L.* 21 (2018): 209.
>
> [17] Dyda, Amalie, et al. "Differential privacy for public health data: An innovative tool to optimize information sharing while protecting data confidentiality." *Patterns* 2.12 (2021).
>
> [18] Rogers, Ryan, et al. "Linkedin’s audience engagements api: a privacy preserving data analytics system at scale." Journal of Privacy and Confidentiality 11 (2021): 3.
>
> [19] Cummings, Rachel, and Deven Desai. "The role of differential privacy in gdpr compliance." *FAT’18: Proceedings of the Conference on Fairness, Accountability, and Transparency*. Vol. 20. 2018.
>
> [20] National Institute of Standards and Technology. (2020, January 27). *Deploying differential privacy for the 2020 Census of population and housing*.
>
> [21] National Institute of Standards and Technology. (2023, December 11). *NIST offers draft guidance on evaluating a privacy protection technique for the AI era*.
>
> [22] National Institute of Standards and Technology. (2025, March 6). *NIST finalizes guidelines for evaluating ‘differential privacy’ guarantees to de-identify data*

---

### Official Review · Reviewer_dic5 · 2025-10-31

**Soundness:** 3
**Presentation:** 3
**Contribution:** 3
**Rating:** 6
**Confidence:** 4

**Summary:**

The paper addresses privacy risks in retrieval-augmented generation when many queries are issued over the same private corpus. It proposes PRIVATE-RAG, which introduces two methods, MURAG and MURAG-ADA, that apply differential privacy at the document level so privacy loss depends on how often each document is retrieved instead of the total number of queries. Experiments on several LLMs and datasets show that the approach maintains good utility while supporting hundreds of queries within a practical privacy budget.

**Strengths:**

Strengths:
1. The paper addresses a relevant and timely problem: how to make RAG systems private when handling many user queries.
2. The idea of tracking privacy at the document level rather than per query is simple but effective, and it clearly reduces the cost of privacy accounting.
3. The two proposed variants are well motivated for different query patterns.
4. The experiments cover a good range of datasets and models, showing solid performance under realistic privacy budgets.

**Weaknesses:**

1. Practical limitation: The proposed method is well-motivated and theoretically sound. However, in realistic deployments there are often scenarios where a single query—or a group of related queries—has high semantic overlap with a large portion of the corpus. In such settings, the thresholding mechanism would struggle to filter out many documents, so most entries would still be “touched” and thus consume privacy budget. This situation can substantially reduce the efficiency advantage of per-document accounting when the retrieval space is broadly relevant.
2. Presentation clarity: The method section is technically solid but somewhat dense; adding an overall flow diagram that shows how retrieval, thresholding, and privacy accounting connect would make the process much easier to follow.

**Questions:**

During adaptive thresholding in MURAG-ADA, the algorithm discretizes similarity scores into bins [a_i,a_i+1). How are the number of bins and boundaries chosen in practice, are they global or per-query, and how does this affect privacy and retrieval precision?

---

> ### Author Response · Authors · 2025-11-27
>
> We thank the reviewer for their positive assessment, for highlighting the importance of the problem, and for acknowledging the strength of our algorithmic design and the comprehensiveness of our empirical evaluation.
>
> > **Weakness 1**: "Practical limitation"
>
> We agree that this issue can occur for the **fixed-threshold** method in Algorithm 1 (MURAG). If a global threshold $\tau$ is used and many documents have relevance scores clustered around $\tau$, then a large fraction of the corpus may enter the active set, which indeed diminishes the efficiency gains from per-document accounting.
>
> To address this, we have introduced Algorithm 2 (MURAG-Ada, marked as Algorithm 5 in the original submission), which replaces the fixed $\tau$ with an **adaptive threshold**. In MURAG-Ada, the threshold is chosen adaptively so that only about $k$ documents per query enter the active set (Lines 3–10 of Algorithm 2). At the same time, the vast majority of the corpus remains untouched. This design preserves the benefits of per-document accounting even when many documents are relevant to a given query. Our empirical results confirm these advantages: on the MQuAKE dataset—a multi-hop question set where (i) each query has semantic overlap with multiple documents and (ii) the sets of relevant documents for different queries also overlap substantially—MURAG-Ada achieves significantly better performance than MURAG (Figure 2, third row).
>
> > **Weakness 2**: "Presentation clarity"
>
> Thanks for your suggestion. We have added a flow diagram on Page 4 to visualize the workflow of our algorithms (marked as Figure A).
>
> > Question: "During adaptive thresholding in MURAG-ADA...and how does this affect privacy and retrieval precision?"
>
> In our experiments, we use a **global uniform grid** over the range of similarity scores, with the same bin boundaries for all queries and datasets. The bin-size is chosen to be $0.2$ and the privacy budget $\varepsilon_{\mathrm{thr}} = 1.0$, which is modest relative to the total budget (e.g., $\varepsilon = 10$).
>
> To further illustrate the privacy-utility trade-off w.r.t. bin sizes, we have conducted an ablation study in the revised version (Figure 5, Appendix F.3). Under $\varepsilon_{\mathrm{thr}} = 1.0$, we are able to achieve an absolute error of $0.2$ w.r.t. the ground truth top-50 relevance score (Figure 5 (Left)). Since the similarity scores mostly lie between $60$ and $80$ (Figure 5 (Right)), this level of perturbation is negligible for retrieval precision. For additional details on the experiment set-up, please refer to Appendix F.3.

---

### Official Review · Reviewer_cR8W · 2025-11-02

**Soundness:** 3
**Presentation:** 3
**Contribution:** 3
**Rating:** 6
**Confidence:** 1

**Summary:**

This paper addresses a critical gap in differentially private (DP) retrieval-augmented generation (RAG): existing DP-RAG methods are limited to single-query settings and become impractical under composition when handling multiple queries. To overcome this, the authors propose MURAG and its adaptive variant MURAG-ADA, two novel multi-query DP-RAG algorithms that leverage per-document individual Rényi privacy filters to track privacy loss based on document retrieval frequency rather than total query count. MURAG uses a fixed relevance threshold to select documents, while MURAG-ADA privately releases query-specific thresholds via noisy prefix sums to better handle correlated queries. The methods are evaluated across three LLMs (OPT-1.3B, Pythia-1.4B, Mistral-7B) and four datasets—including independent (Natural Questions, TriviaQA), correlated (MQuAKE), and privacy-sensitive (ChatDoctor) benchmarks. Results show that both methods achieve meaningful utility under a realistic total privacy budget (ε ≈ 10) for hundreds of queries, significantly outperforming naive composition baselines and synthetic-data alternatives, while also defending against state-of-the-art multi-query membership inference attacks.

**Strengths:**

1. Practical and Timely Problem: The paper tackles a real-world limitation of current DP-RAG systems—scalability to multiple queries—making DP-RAG viable for production deployments in sensitive domains like healthcare or legal services.

2. Comprehensive Evaluation: The experiments span diverse datasets, LLMs, query correlation regimes, and include both utility metrics and robustness against a strong multi-query membership inference attack (Interrogation Attack). The results consistently validate the claims.

**Weaknesses:**

I must candidly note that I am not deeply familiar with the technical nuances of differential privacy (DP), particularly advanced topics such as Rényi DP filters and individual privacy accounting. Consequently, I may have missed potential weaknesses in the paper’s privacy analysis, algorithmic design, or theoretical claims. I recommend that the reviewers with stronger expertise in DP carefully examine the correctness and novelty of the privacy guarantees and the implementation details of the proposed mechanisms.

**Questions:**

see the weakness

---

> ### Author Response · Authors · 2025-11-27
>
> We thank the reviewer for their positive evaluation, for highlighting the importance of the problem, and for recognizing the strength and comprehensiveness of our empirical results.
>
> We believe it is important for the paper to appeal to the broader ICLR audience. To improve readability for non-DP experts, we have revised the presentation of the proofs and added further details in Appendix F.1 and F.2 of the revised version. In addition, we have added a flow diagram on page 4 to better visualize the process of our algorithms. We also hope that the discussion in our general response is clear and that it has helped to highlight our novel technical contributions.
>
> Regarding your question about the Rényi DP (RDP) filter and individual privacy accounting, we provide a brief high-level introduction to these concepts for your reference:
>
> - A **Rényi DP filter** is a mechanism that keeps track of how much privacy budget has been spent as the RAG system answers queries, using the standard RDP accountant. Before answering each new query, the filter checks the current RDP privacy loss against a pre-specified threshold. If the budget is still below the threshold, the query is allowed. Otherwise, further use is blocked. Privacy filters like these are quite tricky for $(\varepsilon,\delta)$-DP (see the work by Ryan et al: https://arxiv.org/abs/1605.08294). However, Zrnic and Feldman have shown that it is rather natural under the RDP — a technique that we adopt in this work.
>
> - **Individual privacy accounting** means that this privacy budget is tracked per individual (or per document), rather than only at the dataset level. In other words, the mechanism maintains a separate privacy loss bound for each document and stops using that document once its budget is exhausted, while other documents with a valid privacy budget remain available for incoming queries. In contrast, under global privacy accounting, a single heavily used document can exhaust the global privacy budget and consequently shut down the entire RAG system.

---

### Author Response · Authors · 2025-11-27
**Change log for the revision**

For the main text (**revisions highlighted in blue**):
* We moved the original Algorithm 5 into the main text to improve readability, where it now appears as Algorithm 2 on page 5.
* We corrected a typo in Line 53 and added the definition of $S \Delta \tilde S$ (Lines 78--79).
* To further improve readability:
(1) We added a flow diagram, Figure A, on Page 4 that visualizes the workflow of our algorithms MURAG and MURAG-Ada.
(2) We rewrote corresponding paragraphs in “Results on privacy-sensitive applications” (lines 412–427) and the related content (lines 455–457).

For the Appendix (all revisions are included in **Appendix F**):
- **F.1 Technical details.** We add the sensitivity calculation of the exponential mechanism in Algorithm 4 (DP-RAG) and provide a more detailed privacy proof for Algorithm 4.
- **F.2 Revised proof.** We rewrite the proof of Theorem 1 (the privacy guarantee of MURAG, previously stated in Appendix D.1) with additional details for clarity.
- **F.3 New ablation experiment (bin size).** We include an ablation study on the choice of bin size in MURAG-Ada (Algorithm 2).
- **F.4 New ablation experiment (threshold).** We include an ablation study on the effect of the threshold $\tau$ in MURAG (Algorithm 1).
- **F.5 New experiment on data extraction attack.** We further evaluate the empirical privacy guarantee of our methods under a data extraction attack.

---

### Author Response · Authors · 2025-11-27
**Clarification of the Technical Novelty**

We clarify and summarize our main technical contributions in response to the related questions raised by Reviewers cR8W and hg6W, as well as making them clearer to the general ICLR audience.

1. **Why use individual privacy accounting and privacy filter rather than amplification by subsampling?**

A key limitation of the previous DP-RAG approach is that privacy loss inevitably grows with the number of queries[1]. Even under advanced composition, the overall privacy budget still increases (albeit sublinearly) as more queries are answered. A standard workaround in classical DP or the DP-LLM literature (eg. [2, 3]) is *privacy amplification by subsampling*, where each query is answered on a randomly subsampled subset of the document corpus. However, this strategy is not suitable for RAG. For many queries, only a small fraction of documents are truly relevant. Additional random subsampling further lowers the probability that those relevant documents are even presented to the DP-RAG mechanism, which in turn degrades answer quality.


2. **Algorithmic technique for incorporating individual privacy accounting into DP-RAG.**

Incorporating individual privacy accounting into DP-RAG is non-trivial. Standard DP-RAG retrieves relevant documents via kNN, i.e., by selecting the top-K documents for a given query. At first glance, one would expect that only the chosen document within the Top-K incurs privacy loss, but that isn’t true: whether a document appears in the top-K depends on all other documents in the corpus, too. This dependence breaks the assumptions required by existing machinery on “individual privacy filter’’ (Zrnic and Feldman).

We solved this problem by introducing a global relevance threshold $\tau$ on the similarity scores: only documents whose scores exceed $\tau$ are ever retrieved and thus incur privacy cost. The Top-K inside the pre-screened list is used for DP-RAG.  Each document is assigned its own privacy budget, and once this budget is exhausted, the document is deterministically excluded from all future retrievals (as stated in Lines 2 and 3 of Algorithm 1).  The prescreening step is critical in decoupling the potential dependence of the Top-K on the rest of the dataset, and ensures that, in practice, only a small number of documents incur new privacy losses per query. We added more details on how this thresholding enables rigorous per-document privacy accounting in Appendix F.2 of the revised version.

Instead, our method adopts *individual privacy accounting* combined with a *privacy filter*. Empirically, we observe that different queries tend to touch different subsets of documents. Under our formulation, the privacy budget is only spent on documents that are actually relevant to a given query, rather than on the entire corpus. This avoids the drawback that the privacy cost must scale with the total number of queries. We further control the overall privacy loss via a document-level (individual) privacy filter that disables a document once its budget is exhausted. This per-document perspective is more natural for multi-query RAG workloads and, crucially, allows us to save substantial privacy budget compared to dataset-level accounting, leading to significantly better utility (see Figure 2).


[1] Koga, Tatsuki, Ruihan Wu, and Kamalika Chaudhuri. "Privacy-Preserving Retrieval-Augmented Generation with Differential Privacy." *arXiv preprint arXiv:2412.04697* (2024).

[2] Wu, Tong, et al. "Privacy-Preserving In-Context Learning for Large Language Models." *The Twelfth International Conference on Learning Representations* (2024).

[3] Hong, Junyuan, et al. "DP-OPT: Make Large Language Model Your Privacy-Preserving Prompt Engineer." *The Twelfth International Conference on Learning Representations* (2024).

---

### Author Response · Authors · 2025-11-27
**Clarification of the Technical Novelty (continued)**

3. **Handling query heterogeneity via adaptive thresholding.**

A single, fixed relevance threshold can be suboptimal because the distribution of relevance scores can shift substantially across queries (see the third paragraph in Section 3.1). For a more adaptive design, one might wish to follow an estimate-then-release scheme and choose a query-dependent threshold via DP quantile estimation. However, standard DP quantile mechanisms operate over the entire dataset, which couples the privacy consumption of all documents and breaks our individual privacy filter: the privacy loss is no longer localized to only those documents actually used for a query.

To address this, we design a noisy prefix-sum–based procedure over relevance scores and provide a novel analysis such that for the privacy accounting of this threshold release, we only need to pay the privacy budget for the documents above this released threshold, which is a generally small set (please refer to Algorithm 2). This design of adaptive thresholding and the analysis aligns well with the individual privacy accounting framework. It significantly improves performance on correlated-query benchmarks such as MQuAKE, as shown in Figure 2.

---

### Meta-Review · Area_Chair_Fcri · 2026-01-03

**Summary:**

This paper focuses on the problem of membership leakage in RAG systems. The authors develop two DP algorithms to provide formal privacy guarantees in multi-query scenarios. While the technical contributions of this work are sound, reviewers have raised concerns primarily regarding its novelty and practical implications. With respect to the first point, the authors have clarified the contributions of their work. However, with respect to the second point, the authors cannot provide a real-world incident demonstrating that RAG systems actually compromise user privacy through membership leakage. Given the security community's current focus on real-world implications, I have to reject this paper. I think it will eventually  be good a work after incorporating practical cases in a revised version.

**Reviewer Concerns:**

#### **Addressed Concerns**

The authors have addressed the issues related to limited experimental evaluation (Reviewer dic5, Reviewer hg6W, Reviewer mpC4), presentation clarity (Reviewer dic5), and sensitivity to parameter tuning (Reviewer mpC4).



#### **Unsolved Concerns**

Nevertheless, the privacy implications remain unresolved (Reviewer hg6W). Specifically, the authors are unable to provide a real-world incident demonstrating that RAG systems actually compromise user privacy through membership inference. Moreover, in the data extraction attacks (Reviewer hg6W), the authors do not measure the leakage rate of PII tokens. Instead, they use ROUGE-L and BERTScore as evaluation metrics, which are not convincing for assessing privacy leakage, as these metrics have seldom been used for this purpose.

**Reviewer Scores:**

I think Reviewer mpC4 might improve his score from 4  to 6, as the authors have adequately addressed the concerns. And others may maintain their original scores.

---

### Decision · Program_Chairs · 2026-01-26

Reject